# Adaptive Rank Control for Robust Reinforcement Learning

**Chenliang Li**[*]                                                                                      *chenliangli@tamu.edu*
**Junyu Leng**[*]                                                                                         *levileng@tamu.edu*
*Texas A&M University*
[*]Equal contribution.

**Jiaxiang Li**                                                                                           *jasonli.optml@gmail.com*
*University of Minnesota*

**Youbang Sun**                                                                                          *ybsun@mail.tsinghua.edu.cn*
*Tsinghua University*

**Shixiang Chen**                                                                                         *shxchen@ustc.edu.cn*
*University of Science and Technology of China*

**Shahin Shahrampour**                                                                                   *s.shahrampour@northeastern.edu*
*Northeastern University*

**Alfredo Garcia**                                                                                        *alfredo.garcia@tamu.edu*
*Texas A&M University*

**Reviewed on OpenReview:** *https://openreview.net/forum?id=lG7VizZQcS*

## Abstract

Robust reinforcement learning (RL) is commonly formulated as a min–max optimization problem to account for epistemic uncertainty in transition dynamics. While theoretically appealing, such formulations are computationally demanding and often induce overly conservative policies. In contrast, we adopt a representation-centric perspective on robustness. Rather than optimizing policies against worst-case transition kernels within a Wasserstein uncertainty set, we study how uncertainty in the dynamics impacts the statistical estimation, and how this sensitivity can be mitigated through structured representations. In particular, we show that low-rank representations provide an implicit mechanism for controlling sensitivity to epistemic uncertainty. In the neural tangent kernel regime, we show that training with uniformly sampled dynamics induces a bias–variance tradeoff, with lower-rank policy representations exhibiting reduced sensitivity to epistemic uncertainty. Within the framework of entropy-regularized RL, we formulate robust representation learning as a bi-level optimization problem that balances expressiveness and robustness. Notably, the resulting solution is conservative at the representation level, even though it does not explicitly optimize a worst-case control objective. We propose an adaptive rank-selection algorithm to approximate a solution to the bi-level formulation, establish policy convergence guarantees, and demonstrate empirically on MuJoCo continuous-control benchmarks that the proposed approach offers a scalable and computationally efficient alternative to classical robust RL. The codes are available here

## 1 Introduction

Reinforcement learning (RL) systems deployed in real-world environments must operate under imperfect knowledge of transition dynamics. In addition to inherent stochasticity (aleatoric uncertainty), practical settings often involve substantial *epistemic uncertainty* arising from limited data, modeling assumptions, or nonstationarity. Such uncertainty is central in applications including robotics, control, environmental policy, and economics (Nagami & Schwager, 2023; Zhou et al., 1996; Lemoine & Traeger, 2014; Hansen & Sargent, 2008). Ignoring epistemic uncertainty can lead to

policies that perform well under nominal models but degrade significantly when the true dynamics deviate from those assumptions.

A standard approach to addressing epistemic uncertainty in RL is *robust policy optimization*, in which the agent seeks a policy that maximizes expected return under worst-case transition dynamics drawn from a prescribed uncertainty set (Iyengar, 2005; Wiesemann et al., 2013; Zhou et al., 1996). While this min–max formulation offers theoretical performance guarantees, it presents two well-known difficulties. First, computing robust Bellman updates requires solving nested inner-loop optimization problems to identify worst-case transitions, which becomes computationally intractable in continuous or high-dimensional domains (Wang & Zou, 2022). Second, worst-case formulations can induce excessive conservatism, yielding policies that sacrifice nominal performance due to pessimistic uncertainty modeling (Mannor et al., 2012; 2016; Xu & Mannor, 2012). Recent work has proposed approximate robust planning (Clavier et al., 2023) and online robust learning schemes (Wang & Zou, 2021), yet the trade-off between robustness, scalability, and expressiveness remains unresolved.

In this work, we adopt an alternative perspective on robustness in reinforcement learning. Rather than optimizing against adversarial transitions, we consider policies trained under transition dynamics *sampled* from an epistemic uncertainty set, and we study how the *complexity of the policy representation* mediates robustness. Our central premise is that robustness to epistemic uncertainty can be achieved by controlling representational capacity, aligning policy complexity with the intrinsic dimension of the task (Li et al., 2018). From this viewpoint, robustness arises through variance reduction induced by constrained expressiveness, rather than worst-case pessimism.

From an epistemic standpoint, sampling transition dynamics from a Wasserstein uncertainty set provides a natural representation of partial knowledge about the environment. Wasserstein balls capture distributional uncertainty in a *metric* sense, allowing nearby models to differ in structured and state-dependent ways rather than through worst-case pointwise deviations. This aligns with epistemic uncertainty arising from finite data, where estimation error induces smooth perturbations of transition distributions rather than adversarial shifts. Uniform sampling over a Wasserstein ball therefore treats uncertainty as a source of stochastic variability to be averaged over, rather than as an adversary to be guarded against. In this sense, Wasserstein sampling induces a form of epistemic regularization: policies are optimized to perform well across a family of plausible models weighted evenly by their proximity to the nominal dynamics, mitigating overconfidence while avoiding the excessive conservatism characteristic of worst-case robust formulations.

We formalize this idea using a bi-level optimization framework. At the lower level, we optimize a policy under transition dynamics sampled from a Wasserstein ball around a nominal model, subject to a fixed low-rank constraint on the policy representation. At the upper level, we adapt the rank to balance robustness and expressiveness. This formulation builds on and extends prior work on low-rank structure in reinforcement learning. In model-based settings, low-rank dynamics have been exploited for joint feature and policy learning (Agarwal et al., 2020; Bose et al., 2024). In model-free settings, the notion of *Bellman rank* has been introduced to characterize the intrinsic complexity of value-function approximation (Jiang et al., 2017), with subsequent efforts aimed at encouraging low Bellman rank during learning (Modi et al., 2024). More recently, Tiwari et al. (2025) showed that reachable state distributions induced by wide neural policies concentrate on low-dimensional manifolds. However, existing work does not address how representational rank should be selected or adapted in the presence of epistemic uncertainty.

**Contributions.** We propose a framework for robust reinforcement learning based on adaptive low-rank policy representations. Our contributions are threefold:

1. We introduce a robustness mechanism based on uniform sampling of transition dynamics from a Wasserstein uncertainty set combined with explicit control of policy rank.
2. We formulate policy rank selection as a bi-level optimization problem that captures the trade-off between robustness (variance reduction) and expressiveness (approximation bias).
3. We develop an alternating optimization algorithm that adapts policy rank during learning and provide empirical evidence on continuous-control benchmarks that adaptive rank selection improves robustness relative to fixed-rank and robust baselines.

**Organization.** Section 2 reviews related literature. Section 3 analyzes the bias–variance trade-off induced by sampling transition dynamics from epistemic uncertainty sets in entropy-regularized RL, establishing that low-rank representations

reduce variance under epistemic uncertainty. Section 4 presents the formulation and the proposed adaptive algorithm. Section 5 reports empirical results on MuJoCo continuous-control benchmarks.

## 2 Preliminary and Related Works

### 2.1 Notation

**Discounted Markov Decision Process.** A Markov decision process (MDP) is represented by the tuple $(\mathcal{S}, \mathcal{A}, P, \rho, r, \gamma)$ wherein $\mathcal{S}$ is the state space $\mathcal{A}$ is the action space (with both $\mathcal{S} \subset \mathbb{R}^n, \mathcal{A} \subset \mathbb{R}^m$ assumed compact), $P_{s,a} \in \Delta_{\mathcal{S}}$, is the transition kernel for $a \in \mathcal{A}, s \in \mathcal{S}$. (where $\Delta_{\mathcal{S}}$ denotes the space of probability measures with support $\mathcal{S}$), $\rho(\cdot)$ is the initial state distribution, $R : \mathcal{S} \times \mathcal{A} \to \mathbb{R}$ is the the reward function $\gamma \in [0, 1)$ is the discount factor. Given $s \in \mathcal{S}$, a policy $\pi$ is a map $\pi(\cdot|s) : \mathcal{S} \to \Delta_{\mathcal{A}}$, where $\Delta_{\mathcal{A}}$ denotes the space of probability measures with support $\mathcal{A}$.

**Epistemic Uncertainty in State Dynamics.** To model uncertainty in the environment dynamics, we introduce an ambiguity set of possible transition kernels:

$$\mathcal{P}_{s,a} := \{P_{s,a} \in \Delta_{\mathcal{S}} \mid W^1(\hat{P}^\circ_{s,a}, P_{s,a}) \leq \epsilon\},$$

where $\hat{P}^\circ_{s,a}$ is a reference transition kernel (e.g., a maximum likelihood estimator obtained from a finite demonstration dataset), $W^1(\hat{P}^\circ_{s,a}, P_{s,a})$ denotes the Wasserstein distance (Villani et al., 2008), and $\epsilon > 0$ is the uncertainty radius. We refer to $P^\circ_{s,a}$ as the centroid of the uncertainty set, representing the true but unobserved transition kernel that governs the system dynamics. Throughout, we assume that epistemic uncertainty is well captured by the Wasserstein ball (Mohajerin Esfahani & Kuhn, 2018), i.e., $P^\circ_{s,a} \in \mathcal{P}_{s,a}$ for all $(s, a) \in \mathcal{S} \times \mathcal{A}$.

**Singular Value Decomposition (SVD)** Let $\theta \in \mathbb{R}^{d_1 \times d_2}$. A *thin* singular value decomposition (SVD) is given by $\theta = \mathbf{U}\mathbf{\Sigma}\mathbf{V}^\top$, where $\mathbf{U}$ is a $d_1 \times r$ matrix with orthogonal columns, that is, an element of the Stiefel manifold (Chakraborty & Vemuri, 2019; Atiyah & Todd, 1960)

$$\mathrm{St}(r, d_1) = \{\mathbf{U} \in \mathbb{R}^{d_1 \times r} : \mathbf{U}^T \mathbf{U} = \mathbf{I}\},$$

$\mathbf{\Sigma}$ is a $r \times r$ diagonal matrix with positive entries $\sigma_1 \geq \sigma_2 \geq \cdots \sigma_r > 0$ (referred to as singular values) and $\mathbf{V} \in \mathrm{St}(r, d_2)$. The singular value decomposition exists for any matrix $\theta \in \mathbb{R}^{d_1 \times d_2}$. We refer to a *truncated* SVD whenever $r < \mathrm{rank}(\theta)$.

### 2.2 Robust Reinforcement Learning

In MDPs, the system dynamics $P$ is usually assumed to be constant over time. However, in the real world, it is subject to perturbations that can significantly impact performance in deployment (Zhang et al., 2023; Moos et al., 2022). Robust MDPs provide a theoretical framework for taking this uncertainty into account, taking $P$ as not fixed but chosen adversarially from an uncertainty set $\mathcal{P}$ (Iyengar, 2005; Nilim & El Ghaoui, 2005), where $\mathcal{P}$ denotes a set of plausible transition models known as the uncertainty set. The objective of robust RL is to find a policy that performs well under the worst-case dynamics within this set. Formally, the robust objective $\mathcal{J}_{\mathcal{P},\pi}$ is defined as:

$$\mathcal{J}_{\mathrm{robust}}(\pi) = \max_\pi \min_{P \in \mathcal{P}} \mathbb{E}_{P,\pi}\Big[\sum_{t \geq 0} \gamma^t R(s_t, a_t) \Big| s_0 \sim \rho_0\Big] \tag{1}$$

The optimal policy $\pi^*_{\mathcal{P}}$ is defined as the solution to the *outer-loop* problem, which maximizes $\mathcal{J}_{\mathrm{robust}}(\pi)$ by accounting for the worst-case transition model at each time step. This leads to the *inner-loop* problem of identifying the worst-case dynamics, for which several approaches have been developed, including value iteration (Nilim & El Ghaoui, 2005; Iyengar, 2005; Wiesemann et al., 2013; Grand-Clément & Kroer, 2021; Kumar et al., 2023a), policy iteration (Kumar et al., 2022; Badrinath & Kalathil, 2021), and policy gradient methods (Li et al., 2022; Wang & Zou, 2022; Wang et al., 2023; Kumar et al., 2023b). However, the problem remains NP-hard for general uncertainty sets, and optimal policies may even be non-stationary (Wiesemann et al., 2013). Most existing methods sidestep this difficulty by assuming

that the inner-loop optimization can be solved efficiently—a reasonable assumption in tabular settings with small uncertainty sets, where one can exhaustively evaluate all transition kernels $P \in \mathcal{P}$. Yet, when the uncertainty set is continuous, the inner-loop problem becomes substantially more challenging and computationally expensive. To address this challenge, Zhou et al. (2023); Gadot et al. (2024) propose the RNAC and EWoK algorithms, which rely on sampling-based techniques to estimate value functions under worst-case dynamics. Although theoretically sound, these methods require drawing multiple next states for each state-action pair, leading to high sample complexity and considerable computational overhead.

## 2.3 Reinforcement Learning with low rank structure

Another direction of research to address this uncertainty is to take advantage of *low-rank structures in dynamics*. In many stochastic control tasks, the transition dynamics admit a low-rank decomposition over a finite set of state-action features (Rozada et al., 2024; 2021; Yang et al., 2019). For example, Tiwari et al. (2025) show that under suitable assumptions, the set of attainable states lies on a low-dimensional manifold. In fixed environments, the dimension of this manifold grows only linearly with the size of the action space and is independent of the state-space dimension. Building on this observation, they employ a $(2d_a + 1)$-dimensional low-rank manifold and apply sparse reinforcement learning methods to solve MuJoCo control tasks. More generally, low-rank structure can be imposed either on the transition kernel or directly on the optimal action-value function $Q^*$, and empirical evidence suggests that $Q^*$ and near-optimal Q-functions in common stochastic control tasks indeed exhibit low-rank properties (Sam et al., 2023; Rozada et al., 2024; 2021; Yang et al., 2019).

Motivated by these findings, algorithms for joint feature and policy learning in *model-based* RL have been developed (Agarwal et al., 2020; Bose et al., 2024), though they typically assume the rank is known a priori. For *model-free* RL, Jiang et al. (2017) introduced the notion of *Bellman rank* to quantify the intrinsic complexity of value function approximation. More recent approaches exploit low-rank factorizations or representations to implicitly encourage small Bellman rank while optimizing the policy or value function (Modi et al., 2024). However, the theoretical guarantees in these works generally rely on fixed dynamics, and to date there is no algorithm that simultaneously recovers the exact Bellman rank while learning the optimal policy under uncertain or time-varying environments.

# 3 Bias–Variance Tradeoff in RL with Epistemic Uncertainty

In this section, we develop a theoretical perspective on robustness in reinforcement learning that departs from worst-case optimization (Nilim & El Ghaoui, 2005). Rather than seeking policies that optimize performance under adversarial transition dynamics which is often computationally prohibitive and overly conservative, we study robustness under epistemic uncertainty modeled via uniformly sampled transition dynamics from an uncertainty set.

Our goal is to understand how uncertainty in the transition model interacts with the *complexity of the policy representation*, and in particular how the *rank* of the model mediates a tradeoff between robustness and expressiveness. We show that, when transition dynamics are sampled from an uncertainty set, restricting the model rank suppresses sensitivity to epistemic perturbations (variance), while introducing approximation error due to limited expressiveness (bias). This induces a principled bias–variance tradeoff with respect to model rank under epistemic uncertainty.

This theoretical analysis serves as the foundation for our algorithmic design in the next section. By characterizing how model rank controls robustness without resorting to worst-case optimization, we obtain guidance for adaptive rank selection in robust reinforcement learning with reduced computational complexity. Our analysis is carried out in the framework of entropy-regularized reinforcement learning (Haarnoja et al., 2018), which admits a convenient variational characterization and yields analytically tractable optimality conditions.

## 3.1 Entropy-Regularized RL under Epistemic Uncertainty

We begin by introducing the entropy-regularized reinforcement learning framework under uniformly sampled transition dynamics, which provides a tractable foundation for the subsequent tradeoff analysis. With policy entropy $\mathcal{H}(\pi(\cdot|s)) :=$

$-\sum_{a \in \mathcal{A}} \pi(a|s) \log \pi(a|s)$, the regularized discounted value with *uncertain dynamics* is defined as:

$$J(\pi) = \mathbb{E}_{a_t \sim \pi, s_{t+1} \sim \mathcal{P}_{s_t, a_t}} \Big[ \sum_{t=0}^{\infty} \gamma^t \big( R(s_t, a_t) + \mathcal{H}(\pi(\cdot|s_t)) \big) \Big],$$

where $s_{t+1} \sim \mathcal{P}_{s_t, a_t}$ denotes the Markov kernel is *sampled uniformly* from the uncertainty set $\mathcal{P}_{s_t, a_t}$. This sampling treats epistemic uncertainty as stochastic variability over plausible models rather than as an adversarial perturbation. For a fixed policy $\pi$, define the soft value functions

$$V^\pi(s) = \mathbb{E}_{\pi, \mathcal{P}} \Big[ \sum_{t \geq 0} \gamma^t \big( R(s_t, a_t) + \mathcal{H}(\pi(\cdot|s_t)) \big) \mid s_0 = s \Big],$$

$$Q^\pi(s, a) = \mathbb{E}_{\pi, \mathcal{P}} \Big[ \sum_{t \geq 0} \gamma^t \big( R(s_t, a_t) + \mathcal{H}(\pi(\cdot|s_t)) \big) \mid s_0 = s, a_0 = a \Big],$$

where we use the short-hand notation $\mathbb{E}_{\pi, \mathcal{P}}$ for expectation with respect to uniformly sampling dynamics from the uncertainty set. The optimal entropy-regularized policy satisfies (Haarnoja et al., 2018)

$$\pi^*(a|s) = \exp \big( Q^*(s, a) - V^*(s) \big),$$

where $Q^*$ is the fixed point of the soft Bellman operator with *uncertain dynamics*:

$$\mathcal{B}Q(s, a) := R(s, a) + \gamma \mathbb{E}_{s' \sim \mathcal{P}_{s, a}} \Big[ \log \sum_{a'} \exp Q(s', a') \Big] \tag{2}$$

and $V^*(s) = \log(\sum_a \exp Q^*(s, a))$.

## 3.2 Bias-Variance Trade-off in Entropy-Regularized RL under Epistemic Uncertainty

In the main result of this section, i.e. Theorem 1, we characterize a bias-variance trade-off in entropy-regularized RL under epistemic uncertainty when policy models are restricted to be of a certain rank. Specifically, in the Neural Tangent Kernel (NTK) regime, restricting the rank of the representation of the soft-Q function amounts to truncating the spectrum of the soft-Bellman operator with uncertain dynamics defined in Eq. (2). We begin with a few preliminary results in the NTK regime.

### 3.2.1 Neural Tangent Kernel (NTK) Regime

We consider an $L$-layer fully-connected neural network $Q_\theta : \mathcal{X} \to \mathbb{R}$ with minimum width $w$ across hidden layers and activation function $\phi : \mathbb{R} \to \mathbb{R}$. Let $\theta \in \mathbb{R}^d$ denote the vector of *all* trainable parameters (all weight matrices and biases), and let $\bar{\theta}$ be the random initialization. For clarity, we use $W = \{W^{(\ell)}\}_{\ell=1}^L$ to denote the collection of layer-wise weight matrices, which constitutes a subset of $\theta$ (together with the biases). When stating initialization and scaling assumptions, we write them layer-wise in terms of $W^{(\ell)}$, while $Q_\theta$ and $\nabla_\theta Q_\theta$ are taken with respect to the full parameter vector $\theta$. We make formal Assumptions 1 – 4 in Appendix B.1, and the proofs of all lemmas are provided in Appendix C.3.2.

**Assumption 1: NTK parameterization and initialization.** For a width-$w$ network, all weight entries satisfy $W_{ij}^{(\ell)} \overset{\text{i.i.d.}}{\sim} \mathcal{N}(0, 1/w)$ for all layers $\ell$, and all bias entries satisfy $b_i^{(\ell)} \overset{\text{i.i.d.}}{\sim} \mathcal{N}(0, 1)$.

**Assumption 2: Smooth activation.** The activation $\phi$ is twice continuously differentiable.

**Assumption 3: Finite input set.** All statements are required only on a finite set $\mathcal{X}$ of inputs.

**Assumption 4: Local parameter neighborhood.** Let $\bar{\theta}$ be the random initialization. The training trajectory is assumed to remain in a ball of radius $R w^{-1/2}$ around $\bar{\theta}$.

**Lemma 1** (NTK Linearization). *Under Assumptions 1–4, there exist constants $C_1, c_1 > 0$ such that with probability at least $1 - C_1 \exp(-c_1 w)$ over the random initialization $\bar{\theta}$, the following holds uniformly for all inputs $x \in \mathcal{X}$ and all $\theta \in \mathcal{B}(\bar{\theta}, R w^{-1/2})$:*

$$Q_\theta(x) = Q_{\bar{\theta}}(x) + \nabla_\theta Q_{\bar{\theta}}(x)^\top (\theta - \bar{\theta}) + \mathcal{R}(\theta), \tag{3}$$

*where the linearization error satisfies $|\mathcal{R}(\theta)| = \mathcal{O}(w^{-1/2})$.*

*Proof sketch.* Consider the second-order Taylor expansion of $Q_\theta(x)$ around $\bar{\theta}$: $\mathcal{R}(\theta) = \frac{1}{2}(\theta - \bar{\theta})^\top H_{\theta(\tau_x)}(\theta - \bar{\theta})$, where $H_{\theta(\tau_x)} := \nabla_\theta^2 Q_{\theta(\tau_x)}(x)$ is the local Hessian at some $\tau_x \in (0, 1)$. Using the $\ell_\infty$ Hessian bound from Lee et al. (2019, Theorem 2.1), the curvature within $\mathcal{B}$ is uniformly suppressed by the width $w$:

$$\sup_{x \in \mathcal{X}, \theta \in \mathcal{B}} \|\nabla_\theta^2 Q_\theta(x)\|_\infty \le C_H(r_1) w^{-1/2},$$

where $C_H(r_1)$ is a curvature constant independent of $w$. Applying the inequality $|u^\top H u| \le \|H\|_\infty \|u\|_1^2$ and the norm relation $\|u\|_1^2 \le d\|u\|_2^2$ with $d = \Theta(w)$, the remainder is bounded by:

$$|\mathcal{R}(\theta)| \le \frac{C_H(r_1)}{2\sqrt{w}} \cdot d \cdot \|\theta - \bar{\theta}\|_2^2 \le \frac{C_H(r_1)}{2\sqrt{w}} \cdot \mathcal{O}(w) \cdot \mathcal{O}(w^{-1}) = \mathcal{O}(w^{-1/2}),$$

where we used the constraint $\|\theta - \bar{\theta}\|_2 \le r_1 w^{-1/2}$. This establishes that the linearization error vanishes as $w \to \infty$. $\square$

This lemma shows that when the neural network is sufficiently wide ($w$ is large), the Q-function behaves like a linear model near its starting point. While deep networks are generally non-linear and complex, the NTK theory guarantees that their curvature (the Hessian) vanishes as the width increases. This means the function surface becomes flatter, allowing us to approximate the change in $Q_\theta(x)$ using only the first-order gradient.

**Lemma 2** (Soft Bellman Error Linearization). *Under Assumptions 1–4, there exist constants $C_2, c_2 > 0$ such that with probability at least $1 - C_2 \exp(-c_2 w)$ over the initialization $\bar{\theta}$, the soft Bellman error satisfies:*

$$\delta(s, a, s'; \theta) = \delta(s, a, s'; \bar{\theta}) + \Psi(s, a, s')^\top(\theta - \bar{\theta}) + \mathcal{O}(w^{-1/2}), \tag{4}$$

*uniformly over all $(s, a, s') \in \mathcal{S} \times \mathcal{A} \times \mathcal{S}$ and all $\theta \in \mathcal{B}(\bar{\theta}, Rw^{-1/2})$, where the feature representation $\Psi(s, a, s')$ is defined as:*

$$\Psi(s, a, s') = \nabla_\theta Q_{\bar{\theta}}(s, a) - \gamma \sum_{a' \in \mathcal{A}} \pi_{\bar{\theta}}(a'|s') \nabla_\theta Q_{\bar{\theta}}(s', a').$$

*Proof sketch.* Apply Lemma 1 to $Q_\theta(s, a)$ and to each coordinate $\{Q_\theta(s', a')\}_{a' \in \mathcal{A}}$, yielding $Q_\theta = Q_{\bar{\theta}} + \nabla Q_{\bar{\theta}}^\top(\theta - \bar{\theta}) + \mathcal{O}(w^{-1/2})$ uniformly. Let $q_\theta(s') = (Q_\theta(s', a'))_{a' \in \mathcal{A}}$ and note that the log-sum-exp map $g(q) = \log \sum_{a'} e^{q_{a'}}$ is smooth with bounded Hessian (on any bounded set, and $\mathcal{A}$ is finite), hence

$$V_\theta(s') = V_{\bar{\theta}}(s') + \nabla_q g(q_{\bar{\theta}}(s'))^\top(q_\theta(s') - q_{\bar{\theta}}(s')) + \mathcal{O}(w^{-1/2}).$$

Since $\nabla_q g(q_{\bar{\theta}}(s')) = \pi_{\bar{\theta}}(\cdot|s')$, substituting the linearization of $q_\theta - q_{\bar{\theta}}$ gives $V_\theta(s') = V_{\bar{\theta}}(s') + \left(\sum_{a'} \pi_{\bar{\theta}}(a'|s') \nabla_\theta Q_{\bar{\theta}}(s', a')\right)^\top(\theta - \bar{\theta}) + \mathcal{O}(w^{-1/2})$. Plugging the expansions of $Q_\theta(s, a)$ and $V_\theta(s')$ into the definition of $\delta$ yields Eq. (4). $\square$

This lemma extends the linearization property from the Q-function to the Soft Bellman error. It shows that the Bellman residual, which is typically a complex, non-linear function of the parameters, can be approximated as a linear transition in the NTK regime. This result allows the non-convex Bellman error minimization to be analyzed as a linear least-squares problem.

### 3.2.2 Soft Bellman Error with Epistemic Uncertainty

We are ready to analyze the soft-Bellman error under epistemic uncertainty about the environment dynamics. For a given ergodic behavioral policy, we will refer to $\mathcal{P}^\circ$ as the *true* steady-state distribution over the transition tuples $(s, a, s')$, with $\mathbb{E}_{(s,a,s') \sim \mathcal{P}^\circ}[\cdot]$ indicating the expectation under this true law. Given an empirical estimate of the transition kernel denoted by $\hat{\mathcal{P}}$, we define the uncertainty set as a Wasserstein ball $\mathcal{B}(\hat{\mathcal{P}}, \varepsilon)$. We will refer to $\mathbb{E}_{(s,a,s') \sim \mathcal{P}}[\cdot]$ as the expectation taken with respect to uniformly sampling dynamics from this uncertainty set.

The minimization of the approximated soft Bellman error can be written as:

$$\min_\theta \mathbb{E}_{(s,a,s') \sim \mathcal{P}^\circ}\left[\left(\Psi(s, a, s')^\top(\theta - \bar{\theta}) + \delta(s, a, s'; \bar{\theta})\right)^2\right].$$

The first order condition can be written as $A_{\mathcal{P}^\circ}(\theta - \bar{\theta}) = b_{\mathcal{P}^\circ}$ where

$$A_{\mathcal{P}^\circ} := \mathbb{E}_{(s,a,s') \sim \mathcal{P}^\circ}\left[\Psi(s, a, s') \Psi(s, a, s')^\top\right] \quad b_{\mathcal{P}^\circ} := -\mathbb{E}_{(s,a,s') \sim \mathcal{P}^\circ}\left[\Psi(s, a, s') \delta(s, a, s'; \bar{\theta})\right]$$

The minimum-norm solution is $\theta^\circ - \bar{\theta} = A^\dagger_{\mathcal{P}^\circ} b_{\mathcal{P}^\circ}$ where $A^\dagger_{\mathcal{P}^\circ}$ is the pseudo-inverse of $A_{\mathcal{P}^\circ}$. Let $\hat{\mathcal{P}}$ be an estimated transition model and consider a Wasserstein ball $\mathcal{B}(\hat{\mathcal{P}}, \varepsilon) := \{\mathcal{P} : W^1(\mathcal{P}, \hat{\mathcal{P}}) \leq \varepsilon\}$, with $\mathcal{P}^\circ \in \mathcal{B}(\hat{\mathcal{P}}, \varepsilon)$. Let $\mathcal{P}$ be drawn (uniformly randomly) from $\mathcal{B}(\hat{\mathcal{P}}, \varepsilon)$. In this case, minimization of approximate soft Bellman error minimization is written as:

$$\min_\theta \ \mathbb{E}_{(s,a,s')\sim\mathcal{P}}\left[\left(\Psi(s,a,s')^\top(\theta - \bar{\theta}) + \delta(s,a,s';\bar{\theta})\right)^2\right]$$

with first order condition:

$$A_\mathcal{P}(\theta - \bar{\theta}) = b_\mathcal{P} \tag{5}$$

where the perturbed Bellman operator and right-hand side are defined as:

$$A_\mathcal{P} := \mathbb{E}_{(s,a,s')\sim\mathcal{P}}\left[\Psi(s,a,s')\,\Psi(s,a,s')^\top\right], \qquad b_\mathcal{P} := -\mathbb{E}_{(s,a,s')\sim\mathcal{P}}\left[\Psi(s,a,s')\,\bar{\delta}(s,a,s')\right],$$

Consider a truncated singular value decomposition:

$$A_{\mathcal{P},r} = U\Sigma_{\mathcal{P},r}V^\top, \qquad \Sigma_{\mathcal{P},r} = \mathrm{diag}(\sigma_{\mathcal{P},1}, \ldots, \sigma_{\mathcal{P},r}, 0, \ldots, 0),$$

with singular values $\sigma_{\mathcal{P},1} \geq \cdots \geq \sigma_{\mathcal{P},r} > 0$. The truncated solution to (Eq. (5)) is:

$$\theta_r - \bar{\theta} = A^\dagger_{\mathcal{P},r}b_\mathcal{P} \tag{6}$$

We are now ready to formalize the tradeoff between robustness and expressiveness induced by rank-constrained learning under epistemic uncertainty. Before presenting the main result, we establish the core notations. Let $\theta^\circ$ denote the ground-truth optimal parameter and $\theta_r$ represent the estimated parameter derived from a rank-$r$ truncation. We define $A_\mathcal{P}$ and $b_\mathcal{P}$ as the design matrix and target vector under a transition distribution $\mathcal{P}$, where $A_\mathcal{P}$ has a full rank of $d$. In the spectral domain, $v_i$ denotes the $i$-th right singular vector of $A_\mathcal{P}$, and $\sigma_{\mathcal{P},r}$ is its $r$-th singular value. To capture epistemic uncertainty, we denote the Wasserstein ambiguity radius as $\varepsilon$. Furthermore, we assume $A_\mathcal{P}$ and $b_\mathcal{P}$ are Lipschitz continuous with respect to $\mathcal{P}$, with Lipschitz constants $L$ and $L_b$, respectively.

The following result characterizes how perturbations in the transition distribution propagate through the Bellman operator and interact with the low-rank approximation of the policy parameters. Specifically, the theorem decomposes the error $\|\theta_r - \theta^\circ\|_2$ incurred by a rank-$r$ estimator into a variance term driven by epistemic uncertainty in the dynamics and a bias term arising from the restriction of the parameter space. This decomposition makes explicit how the singular value spectrum of the Bellman operator governs the sensitivity to uncertainty, thereby providing a principled basis for adaptive rank selection.

**Theorem 1.** *Bias-Variance Trade-off of Rank-r Approximation: Under Assumptions 1–5:*

$$\|\theta_r - \theta^\circ\|_2 \ \leq \ \underbrace{\frac{2\varepsilon(L_b + L\|\theta^\circ\|_2)}{\sigma_{\mathcal{P},r}}}_{variance} + \underbrace{\|\sum_{i=r+1}^{d} v_i v_i^\top \theta^\circ\|_2}_{bias} \tag{7}$$

*Proof.* We refer to Appendix C.3 for the proof. □

## 3.3 Illustration of Theorem 1

This section provides empirical illustration of the bias–variance trade-off characterized in Theorem 1 Recall that Theorem 1 establishes an explicit decomposition of the parameter error induced by rank-$r$ truncation under epistemic uncertainty: increasing the representation rank reduces approximation bias by retaining more spectral components of the Bellman operator, while simultaneously amplifying variance through sensitivity to perturbations in the transition dynamics, as quantified by the inverse singular values $\sigma_{P,r}^{-1}$.

Since the true parameter error $\|\theta_r - \theta^\circ\|_2$ is not directly observable in reinforcement learning, we validate this trade-off indirectly through policy-level performance. We consider two complementary settings. First, we study a linearized CartPole system with controlled epistemic uncertainty, where the Wasserstein ambiguity set can be characterized

explicitly and the nominal system rank is known. This example isolates the effect of rank selection in a setting closely aligned with the assumptions of our analysis. Second, we extend the validation to nonlinear MuJoCo control tasks, demonstrating that the same qualitative trade-off persists beyond the linearized regime and supports the applicability of adaptive rank selection in practice.

**Example 1: Cartpole with Uncertain Pole Length** In this example, we provide a numerical CartPole example to illustrate the theorem 1. Following the corrected dynamics in Florian (2007), the CartPole system can be represented as a linear transition model with intrinsic rank 4. To introduce epistemic uncertainty and evaluate robustness, we allow the pole length $l$ to vary across episodes. Additional implementation details are provided in Section A.1.

**Validation** Following the setup in Section 4.1, we conduct a numerical experiment to examine whether the model rank affects the performance of this linear control system. We perform a sanity check using models of different ranks. As shown in Figure 1, although the nominal CartPole dynamics suggest an optimal rank of 4 (Eq. (17) in Appendix), introducing model uncertainty requires greater capacity, and the model with rank = 8 achieves the best performance, which is closely aligning with our theoretical prediction.

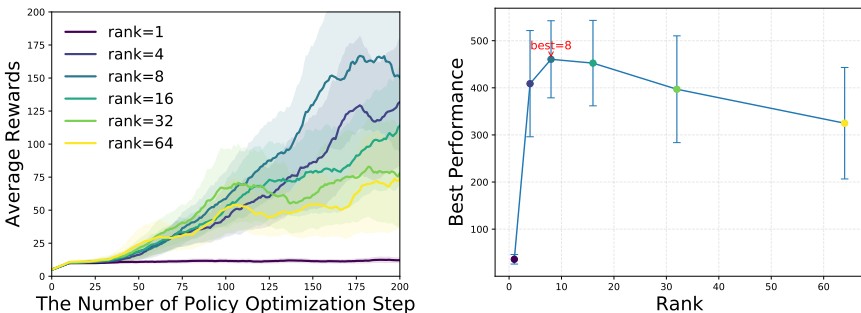

Figure 1: Performance of different-rank policy models under CartPole dynamics uncertainty.

**Example 2: MuJoCo Benchmark Illustration.** To further illustrate the bias–variance trade-off, we consider more complex control tasks in the MuJoCo benchmark (Todorov et al., 2012). Specifically, we employ a three-layer neural network and adopt a rank-control mechanism similar to (Hu et al., 2022; Xu et al., 2019) (see details in Sec.4.1). Our experiments reveal a clear bias–variance tradeoff, as illustrated in Figure 2: models with extremely low-rank representations exhibit high bias, while high-rank models suffer from large approximation errors due to transition samples drawn from uncertain dynamics.

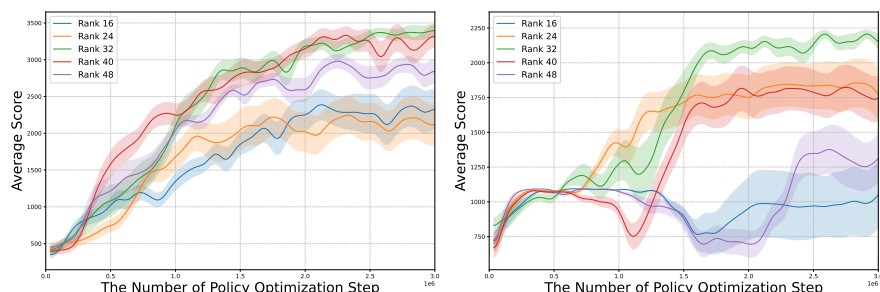

Figure 2: Performance of policy models under high model uncertainty in Walker2d-v3 (**Left**) and Hopper-v3 (**Right**).

# 4 Adaptive Rank Control for Robust Reinforcement Learning

## 4.1 A Bi-level Optimization Formulation

The analysis in previous Section highlights that selecting the policy rank involves a delicate balance: too small a rank induces bias, while too large a rank amplifies variance. This trade-off suggests the need for an adaptive mechanism that can automatically adjust the rank during learning. Motivated by this insight, we introduce a bi-level (Colson et al.,

2007) optimization formulation, where the lower-level problem identifies the optimal policy with uniformly sampled environment dynamics (from a Wasserstein ball around a centroid model) under a fixed rank, and the upper-level problem searches for the representation that optimizes a measure of fit to the lower-level model while regularizing by rank. We consider a parameterized policy $\pi_W$, where $W \in \mathbb{R}^{d_1 \times d_2}$ with $d_1, d_2 > 0$. And we respectively denote by

$$\mathcal{M}_r := \{W \in \mathbb{R}^{d_1 \times d_2} \mid \mathrm{rank}(W) = r\} \quad \mathcal{M}_{\leq \bar{r}} := \{W \in \mathbb{R}^{d_1 \times d_2} \mid \mathrm{rank}(W) \leq \bar{r}\}$$

the smooth manifold of matrices with rank $r$ and the algebraic variety of matrices with rank less than or equal to $\bar{r} > 0$.

**Formulation:** Towards developing an approach that simultaneously learns the policy and low rank representation, we introduce the following bi-level formulation:

$$\min_r \mathbb{E}_{(s,a) \sim \mathcal{P}_{\pi_{W^*}}} \|\mathrm{Proj}_{\mathcal{M}_r}(\pi_{W^*})(a|s) - \pi_{W^*}(a|s)\|_2 + \lambda r \tag{8}$$

$$\text{s.t. } W^* := \arg \max_{W \in \mathcal{M}_{\leq \bar{r}}} \mathbb{E}_{\tau \sim \mathcal{P}_{\pi_W}} \left[ \sum_{t \geq 0} \gamma^t \Big( R(s_t, a_t) + \mathcal{H}(\pi_W(\cdot|s_t)) \Big) \right] \tag{9}$$

where $\mathcal{P}_{\pi_{W^*}}$ denotes the steady-state distribution obtained by uniformly sampling the transition kernel from the Wasserstein ball and selecting actions according to the policy $\pi_{W^*}$, the operator $\mathrm{Proj}_{\mathcal{M}_r}(\pi_{W^*})$ denotes the projection of the policy onto the low-rank manifold $\mathcal{M}_r$, $\lambda$ serves as a weight for rank regularization $r$, where $r$ denotes the rank variable, and $\bar{r}$ represents its maximum allowable value.

**Discussion** The bi-level formulation in Eq. (8)– Eq. (9) plays two complementary roles. The *lower-level problem* Eq. (9) optimizes the policy parameters under a fixed rank constraint, aiming to maximize the entropy-regularized return and thus capture the best achievable policy representation at that rank. However, the optimal solution $\pi_{W^*}$ of the lower-level problem may not align with the intrinsic task complexity and can overfit by exploiting the full representation power. To address this, the *upper-level problem* Eq. (8) explicitly searches for an appropriate rank that balances bias and variance, as motivated in the previous section. It seeks the best low-dimensional representation (bounded by $\bar{r} > 0$) of the state–action value associated with $\pi_{W^*}$, while controlling model capacity through the rank regularization term. In this way, the upper-level problem enforces a bias–variance tradeoff, ensuring that the learned representation achieves robustness without unnecessary over-parameterization.

## 4.2 Algorithm

We are now ready to design algorithms for the proposed formulation. Note that our formulation has a hierarchical structure and falls into the class of bi-level optimization problems (Hong et al., 2023; Colson et al., 2007). In general, bi-level problems are challenging to solve; in our case, the upper-level objective Eq. (8) depends explicitly on the optimal solution of the lower-level problem. Furthermore, the rank regularizer $C(\cdot)$ is non-differentiable, which precludes the use of (stochastic) first-order methods for the upper-level optimization. Fortunately, as we will show, a simple yet effective adaptive greedy search algorithm can be employed to obtain an empirical solution to the upper-level problem. At a high level, the proposed algorithm alternates between two steps: (i) a **Rank Adaptation Step**, which updates the rank $r$ via a greedy search procedure, and (ii) a **Policy Optimization Step**, which optimizes the parameters under the rank constraint $\eta \in \mathcal{M}_{\leq r}$. We now examine each step in detail.

**Rank Adaptation Step** From the discussion in Section 3, we know that extremely low-rank models are limited in their representation power and thus fail to capture sufficient information under model uncertainty. In contrast, high-rank models tend to overfit, resulting in poor generalization. Hence, it is crucial to carefully select an appropriate rank for policies in MDPs with uncertain dynamics. Although Theorem 1 provides useful insights, in practice it is difficult to explicitly solve this tradeoff and obtain the optimal rank. To address this, we adopt a greedy strategy: starting from a high-rank model, we gradually reduce the rank until reaching a stable value that yields consistent performance under model uncertainty. This procedure operationalizes the bias–variance tradeoff characterized in Theorem 1 and forms the core of the **Rank Adaptation Step** in our algorithm.

Specifically, the upper-level problem Eq. (8) requires us to identify suitable representations for both the policy and value models while keeping their ranks as low as possible. If no lower-rank model with sufficient approximation quality can be found, we simply retain the previous rank, i.e., $r_{\mathrm{new}} = r_{\mathrm{old}}$. To do the greedy search, we consider using the following criterion to decide the new rank $\hat{r}$. Note there are many ways to decide the target rank; in the ablation study,

we show that using this criterion achieves a smooth truncation and makes the rank converge to the intrinsic rank of the environment. In Appendix D.2, we show how to impose the rank constraint on policy and value models.

$$\hat{r} = \max\{\ell \in \{1, 2, \ldots, d\} : \frac{\sum_{i=1}^{\ell} \sigma_i}{\sum_{i=1}^{d} \sigma_i} \leq \beta\} \tag{10}$$

**Policy Optimization Step** One can adopt the standard approaches, such as the well-known soft actor critic (SAC) (Haarnoja et al., 2018) algorithm to obtain an approximate optimal policy that solves Eq. (9). Notice that after reconstructing the neural network, the rank of policy parameters $W$ is no larger than $\hat{r}$ due to the existence of the intermediate layer. In this way, the rank constraint is automatically enforced during optimization without requiring explicit SVD at every update. We summarize the proposed algorithm in Algorithm 1, corresponding to the rank adaptation step and policy improvement step.

### 4.3 Convergence of Proposed Algorithm

We formalize the convergence properties of our adaptive low-rank framework by analyzing the interplay between structural stabilization and stochastic optimization. The motivation for this analysis comes from a fundamental challenge in robust RL: balancing the expressiveness of neural representations against the potential for overfitting under epistemic uncertainty. Our primary intuition is that the monotonic rank-adaptation rule ensures the network architecture settles into a deterministic low-rank manifold in finite time, allowing for convergence analysis.

Specifically, our theoretical framework decomposes the convergence into two distinct phases. In the initial phase, the truncation rule Eq. (10) progressively achieves the stationary representation bottleneck $k^*$. Once the rank is stabilized, the SVD-induced projection ceases to introduce structural bias, acting as an identity mapping on the parameter manifold. By integrating these results with the approximation guarantees of multi-layer networks, we prove that the weight trajectory enters a predictable $\epsilon_W$-neighborhood of the stationary point and remains there almost surely. This transition from structural adaptability to asymptotic stability provides a rigorous theoretical foundation for the strong empirical performance observed in our experiments. The proofs of all lemmas are provided in Appendix C.3.2.

**Lemma 3** (Finite-time Rank Stabilization). *Under the adaptive energy rule with threshold Eq. (10) with $\beta \in (0, 1)$, define $k_t$ as the truncation rank selected at iteration $t$, there exists $T_0 < \infty$ such that $k_t = k^*$ for all $t \geq T_0$.*

*Proof Sketch.* By the adaptive energy rule, $k_{t+1} \leq k_t$ for all $t$ and $k_t \in \mathbb{Z}_+$. Thus $\{k_t\}$ is a nonincreasing sequence of positive integers, so it can decrease only finitely many times. Hence there exist $T_0 < \infty$ and $k^* \in \mathbb{Z}_+$ such that $k_t = k^*$ for all $t \geq T_0$. $\qquad\square$

This lemma establishes the *structural consistency* of our algorithm. By establishing that the rank stabilizes in finite time, we ensure that the underlying parameter space remains consistent. This allows us to theoretically characterize the latter stage of training as an optimization process over a fixed-dimensional manifold.

**Lemma 4** (Zero Projection Perturbation). *For $t \geq T_0$, the parameter perturbation induced by the rank-$k^*$ SVD projection is zero: $\|W_t - \Pi_{k^*}(W_t)\|_F = 0$.*

*Proof Sketch.* For $t \geq T_0$, $\text{rank}(W_t) \leq k^*$. By the Eckart-Young-Mirsky Theorem, Eckart & Young (1936), $\|W - \Pi_{k^*}(W)\|_F^2 = \sum_{i=k^*+1}^{d} \sigma_i^2(W)$. Since $\sigma_i = 0$ for $i > k^*$, the projection error vanishes identically, making $\Pi_{k^*}$ an identity mapping on the stabilized manifold. $\qquad\square$

This result confirms that once the bottleneck dimension $k^*$ is identified, the SVD projection stops to introduce projection noise.

**Lemma 5** (Neighborhood Convergence). *By standard Neural Actor-Critic Tian et al. (2023), there exists $T_1 \geq T_0$ and $\epsilon_W > 0$ such that for $t \geq T_1$, the expected weight trajectory satisfies $\mathbb{E}[\|W_t - W^*\|_F] \leq \epsilon_W$.*

*Proof Sketch.* After stabilization ($t \geq T_0$), the system reduces to a fixed-structure AC algorithm. Following Tian et al. (2023), the suboptimality gap $\Delta Q$ is asymptotically bounded by $\epsilon_{gap} = \mathcal{O}(\epsilon) + \tilde{\mathcal{O}}(1/\sqrt{m})$. By Assumption 6 and apply Jensen's Inequality, we obtain the parameter-space bound:

$$\limsup_{t \to \infty} \mathbb{E}[\|W_t - W^*\|_F] \leq \sqrt{2\epsilon_{gap}/\mu} := \epsilon_W \tag{11}$$

This establishes the convergence to an $\epsilon_W$-neighborhood of the stationary point $W^*$. □

Theorem 3.1 of Tian et al. (2023) provides convergence guarantees at the *function level* (e.g., a bounded critic suboptimality $\Delta Q$ for the induced $Q$-function). Assumption 6 serves as the bridge that converts this functional guarantee into a *parameter-space* guarantee: once $\Delta Q$ is small, local strong convexity forces $W_t$ to remain close to $W^\star$ in Frobenius norm. In other words, $Q$-stability implies weight stability in a neighborhood, which is the key step enabling our subsequent approximation arguments.

**Theorem 2** (Almost Sure Convergence of Network Parameters). *The output sequence of the network parameters $\{W_k\}$ converges to a stable parameter configuration $\hat{W}$ with probability 1 (almost surely), where the limit point $\hat{W}$ resides within the $\epsilon_W$-neighborhood of the stationary point $W^*$.*

*Proof.* Let $\mathcal{F}_k = \sigma(W_0, W^1, \ldots, W_k)$ denote the filtration generated by the stochastic parameter sequence. For $k \geq \max(T_0, T_1)$, Lemma 4 guarantees that $\text{rank}(W_k) \leq k^*$, implying $\|W_k - \Pi_{k^*}(W_k)\|_F = 0$. Consequently, the projection operator reduces to an identity mapping, and the discrete-time Actor-Critic update simplifies to an unprojected stochastic gradient step:

$$W_{k+1} = W_k - \alpha_k g_k \tag{12}$$

where $g_k$ is the stochastic gradient estimator. Expanding the squared Frobenius distance to the stationary point $W^*$ yields:

$$\|W_{k+1} - W^*\|_F^2 = \|W_k - W^*\|_F^2 - 2\alpha_k \langle W_k - W^*, g_k \rangle + \alpha_k^2 \|g_k\|_F^2 \tag{13}$$

Taking the conditional expectation with respect to $\mathcal{F}_k$, and assuming the stochastic gradient possesses a uniformly bounded second moment $\mathbb{E}[\|g_k\|_F^2 \mid \mathcal{F}_k] \leq G^2$, we obtain:

$$\mathbb{E}\left[\|W_{k+1} - W^*\|_F^2 \mid \mathcal{F}_k\right] \leq \|W_k - W^*\|_F^2 - 2\alpha_k \langle W_k - W^*, \mathbb{E}[g_k \mid \mathcal{F}_k]\rangle + \alpha_k^2 G^2 \tag{14}$$

By Lemma 5, the optimization trajectory is asymptotically restricted to a bounded region where the expected gradient provides a valid descent direction, ensuring $\langle W_k - W^*, \mathbb{E}[g_k \mid \mathcal{F}_k]\rangle \geq 0$. Defining the Lyapunov sequence $V_k = \|W_k - W^*\|_F^2$, Eq. (14) establishes a non-negative near-supermartingale:

$$\mathbb{E}[V_{k+1} \mid \mathcal{F}_k] \leq V_k + \alpha_k^2 G^2 \tag{15}$$

By assumption, the learning rate sequence satisfies $\sum_{k=1}^\infty \alpha_k = \infty$ and $\sum_{k=1}^\infty \alpha_k^2 < \infty$. Therefore, the variance perturbation term is summable: $\sum_{k=1}^\infty \alpha_k^2 G^2 < \infty$.

By the Robbins-Siegmund Theorem Neri & Powell (2024), the summability of the noise term $\sum \alpha_k^2 G^2 < \infty$ ensures that $V_k \xrightarrow{a.s.} V_\infty < \infty$. Combined with the asymptotic expectation bound from Lemma 5, the sequence $\{W_k\}$ is prevented from oscillating or diverging, converging almost surely to a limit point $\hat{W}$ within the $\epsilon_W$-neighborhood of $W^*$:

$$\mathbb{P}\left(\lim_{k \to \infty} W_k = \hat{W}\right) = 1, \quad \text{where} \quad |\hat{W} - W^*|_F \leq \epsilon_W \tag{16}$$

This concludes the proof of Theorem 2. □

# 5 Experimental Tests

In this section, we present numerical evaluations of the proposed method AdaRL (Alg. 1) and compare it against several robust RL baselines, including RNAC, Parseval regularization, fixed-rank SAC, and the algorithm from Tiwari et al. (2025). Our experiments highlight the advantages of AdaRL in two key aspects: (1) it achieves a favorable trade-off between the bias and variance induced by model uncertainty, thereby enabling more robust policy learning; and (2) it identifies a suitable low-rank manifold, within which constraining the policy model yields a representation that remains robust under model uncertainty. More details are given in the Appendix D.1.

We focus on robotic control tasks with continuous action spaces, using four widely adopted OpenAI Gym environments and their variants: `Hopper-v3`, `Walker2d-v3`, `Ant-v3`, and `Humanoid-v3`. Following the setup in Luo et al.

---

**Algorithm 1** *Adaptive Rank Representation (AdaRL)*

---

1: **Input:** Initialize parameters: state-action value parameters $\theta^0$ and policy parameters $W^0$. Truncation threshold $\beta \in (0, 1)$, and truncate interval $d_t$.
2: **for** $k = 0, 1, \ldots, K - 1$ **do**
3:     **Data Sampling:** Sample trajectories $\tau_1, \ldots, \tau_N$ from the current policy $\pi_{W^k}$, and add them to the replay buffer: $D \leftarrow D \cup \{\tau_1, \ldots, \tau_N\}$.
4:     **Policy Evaluation:** Compute $Q_{\theta^k}(\cdot, \cdot)$ with sampled data $D$.
5:     **Policy Improvement:** $W^{k+1} = W^k - \alpha_k \nabla_W J(W^k; \theta^k)$.
6:     **Rank Adaptation Step:** If $k \mod d_t = 0$, search the suitable rank by Eq. (10) and project $W^k$ into a lower-rank manifold $\mathcal{M}_{\hat{r}}$.
7: **end for**

---

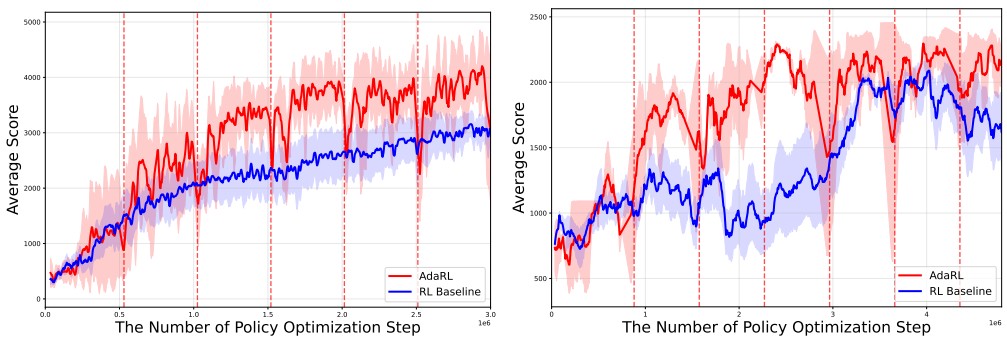

Figure 3: Training performance on MuJoCo tasks. Left:Walker-V3, Right: Hopper-V3
The red dashed vertical lines indicate the boundaries between different iteration intervals.

(2024), we introduce model uncertainty by modifying the source dynamics for each task. In Hopper and Walker2d, this involves structural changes such as adjusting torso and foot sizes, while in Ant and Humanoid we alter physical parameters including gravity or add external forces such as wind with a specified velocity. During training, the environment dynamics vary across episodes to simulate epistemic uncertainty. The baselines considered in this scenario are: (1) SAC (Haarnoja et al., 2018) with a fixed-rank parameterization; (2) RNAC-DS (Zhou et al., 2023), which employs double sampling within newly defined uncertainty sets and uses function approximation to solve the robust Bellman equation; We also evaluate the RNAC-IPM variant to ensure a more comprehensive and fair comparison. (3) Parseval regularization (Chung et al., 2024), which enforces orthogonality in weight matrices to preserve optimization properties and improve training stability in continual reinforcement learning; and (4) the method of Tiwari et al. (2025), which incorporates a fully connected sparsification MLP layer for reinforcement learning.

In Figure 3 and Table 1, we report numerical results comparing the proposed AdaRL algorithm with several baselines. As shown in Figure 3, both AdaRL and standard SAC achieve similar performance in the first iteration; however, once the model rank is adjusted, AdaRL consistently outperforms the standard methods by mitigating the impact of model uncertainty. We further conduct ablation studies to examine the impact of key design components, including the adaptive rank mechanism and replay buffer warm-start strategy in Appendix D.5.3. The results show that the components contribute significantly to performance gains and training stability. In Table 1, the results show that AdaRL consistently outperforms the baselines by a significant margin in most scenarios. As discussed in Section 2.2, robust RL algorithms typically perform policy improvement based on worst-case value functions, which enhances robustness but often yields overly conservative policies and incurs high approximation errors in continuous control environments (Mannor et al., 2012; 2016; Xu & Mannor, 2012). For regularization-based approaches, Parseval regularization can partially mitigate value-function overfitting, but it remains less effective than the low-rank constraint imposed in AdaRL. To fairly assess policy generalization, all evaluations are conducted under the fixed nominal dynamics $\mathcal{P}^\circ$, enabling us to examine whether the learned policies remain effective and robust in the presence of model uncertainty. In Appendix D.5, we further demonstrate the robustness of the trained policy in different perturbed dynamics.

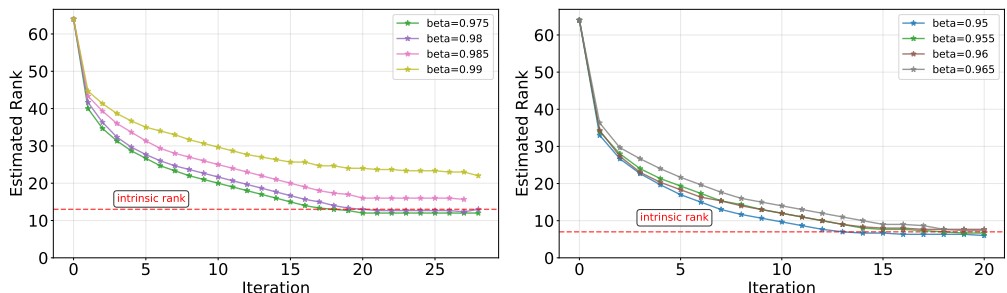

Figure 4: We plot the estimated rank from AdaRL throughout training. The intrinsic rank refers to the value identified by Tiwari et al. (2025). **Left:** Walker2d. **Right:** Hopper.

| Task | AdaRL (proposed) | RNAC-DS | RNAC-IPM | Parseval | Alg. in Tiwari et al. (2025) |
|---|---|---|---|---|---|
| Hopper | $\mathbf{2109.8 \pm 322.90}$ | $1542.36 \pm 62.73$ | $1666.33 \pm 495.82$ | $1410.64 \pm 456.52$ | $1850.09 \pm 234.98$ |
| Walker | $\mathbf{3991.90 \pm 567.00}$ | $1906.68 \pm 620.90$ | $2725.42 \pm 570.29$ | $2368.26 \pm 1346.67$ | $3280.55 \pm 179.42$ |
| Ant | $\mathbf{3067.13 \pm 111.55}$ | $1021.97 \pm 230.71$ | $1827.77 \pm 237.64$ | $2063.18 \pm 381.27$ | $2719.95 \pm 225.11$ |
| Humanoid | $\mathbf{5428.72 \pm 50.10}$ | $2351.42 \pm 443.12$ | $3321.49 \pm 342.31$ | $458.35 \pm 76.41$ | $5255.03 \pm 757.45$ |

Table 1: **MuJoCo Results.** The performance of the benchmark algorithms. Bolded numbers indicate the best results among AdaRL, RNAC-DS, RNAC-IPM, Parseval regularization, and the algorithm in Tiwari et al. (2025) for each task.

In Figure 4, we report an additional experiment showing that the rank estimated by the AdaRL algorithm in Eq. (8) gradually converges to the intrinsic rank identified by Tiwari et al. (2025), given an appropriate choice of $\beta$ in Alg. 1 (set to 0.98 in our experiments). This result demonstrates that AdaRL can effectively search for a suitable rank.

## 6 Conclusion

In this paper, we propose a reinforcement learning framework under epistemic uncertainty based on low-rank representations. We first characterize a bias–variance trade-off induced by low-rank approximation under uncertain dynamics, and then develop the proposed algorithm, which dynamically adjusts the policy rank to balance generalization and robustness.

## Acknowledgments

This work was supported in part by NSF under Award ECCS-2240788 and Award ECCS-2240789.

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

# A Appendix: Experiment Details

## A.1 Cartpole Derivation

Following the corrected dynamics in Florian (2007), the CartPole system can be written as

$$
\begin{bmatrix} \dot{x} \\ \ddot{x} \\ \dot{\theta} \\ \ddot{\theta} \end{bmatrix} = \begin{bmatrix} 0 & 1 & 0 & 0 \\ 0 & 0 & -\frac{mg}{(m+M)\left(\frac{4}{3}-\frac{m}{m+M}\right)} & 0 \\ 0 & 0 & 0 & 1 \\ 0 & 0 & \frac{g}{l\left(\frac{4}{3}-\frac{m}{m+M}\right)} & 0 \end{bmatrix} \begin{bmatrix} x \\ \dot{x} \\ \theta \\ \dot{\theta} \end{bmatrix} + \begin{bmatrix} 0 \\ \frac{1}{m+M} + \frac{m}{(m+M)^2\left(\frac{4}{3}-\frac{m}{m+M}\right)} \\ 0 \\ -\frac{1}{l(m+M)\left(\frac{4}{3}-\frac{m}{m+M}\right)} \end{bmatrix} u,
\tag{17}
$$

where $x$ denotes the horizontal position of the cart, $\dot{x}$ its velocity, $\theta$ the pole angle (measured from the upright position), and $\dot{\theta}$ the angular velocity. The parameters $m$ and $M$ are the pole and cart masses, respectively, $l$ is the pole length (0.5 in the default setting), $g$ is the gravitational acceleration, and $u \in \{0, 1\}$ represents the control input (horizontal force) applied to the cart. To introduce epistemic uncertainty and evaluate robustness, we allow the pole length $l$ to vary across episodes. Further implementation details are provided in Section A.1.

To inject robustness and model uncertainty into this system, we assume that the pole length $l$ varies across episodes. Let $l_0$ denote the nominal length used to define the reference dynamics. During training, we randomly sample $l$ from the interval

$$l \in [0.95\, l_0,\ 1.05\, l_0].$$

For a fixed pole length $l$, the CartPole dynamics in Eq. (17) can be written compactly as

$$\dot{s} = A(l)\, s + B(l)\, u, \qquad s = [x, \dot{x}, \theta, \dot{\theta}]^\top,\ u \in \{0, 1\},$$

where $A(l)$ and $B(l)$ are obtained directly from the matrices in Eq. (17).

Let $l_0$ be the nominal length and denote by

$$s_0^+ = A(l_0)s + B(l_0)u, \qquad s^+(l) = A(l)s + B(l)u$$

the next states under $l_0$ and $l$, respectively. Since the system is deterministic, the transition kernels are Dirac measures

$$\hat{P}_{s,a}^\circ = \delta_{s_0^+}, \qquad P_{s,a}(l) = \delta_{s^+(l)}.$$

For a Wasserstein distance with Euclidean ground cost, we then have

$$W\big(\hat{P}_{s,a}^\circ, P_{s,a}(l)\big) = \big\|s_0^+ - s^+(l)\big\|_2 = \big\|(A(l_0) - A(l))s + (B(l_0) - B(l))u\big\|_2.$$

Assume the state is bounded, $\|s\| \le \|s\|_{\max}$, and recall that $u \in \{0, 1\}$, hence $|u| \le 1$. Using operator norms, we obtain

$$W\big(\hat{P}_{s,a}^\circ, P_{s,a}(l)\big) \ \le\ \|A(l_0) - A(l)\|\, \|s\|_{\max} + \|B(l_0) - B(l)\|.$$

The entries of $A(l)$ and $B(l)$ depend on $l$ only through rational functions such as $\frac{1}{l}$ and $\frac{1}{l(m+M)}$. On the compact interval $l \in [0.95\, l_0, 1.05\, l_0]$ these functions are Lipschitz, so there exist constants $K_A, K_B > 0$ such that

$$\|A(l_0) - A(l)\| \le K_A|l - l_0|, \qquad \|B(l_0) - B(l)\| \le K_B|l - l_0|.$$

Hence, for any $(s, a)$ and any $l \in [0.95\, l_0, 1.05\, l_0]$,

$$W\big(\hat{P}_{s,a}^\circ, P_{s,a}(l)\big) \le \big(K_A\|s\|_{\max} + K_B\big)\, |l - l_0| \le 0.05\, l_0 \big(K_A\|s\|_{\max} + K_B\big).$$

Therefore, by choosing $\varepsilon_{s,a} := 0.05\, l_0 \big(K_A\|s\|_{\max} + K_B\big)$, (or a global $\varepsilon$ using the supremum over $(s, a)$), the perturbed dynamics with $l \in [0.95\, l_0, 1.05\, l_0]$ indeed satisfy

$$P_{s,a}(l) \in \Big\{P_{s,a} \in \Delta_{\mathcal{S}} \ \Big|\ W\big(\hat{P}_{s,a}^\circ, P_{s,a}\big) \le \varepsilon_{s,a}\Big\},$$

i.e., they lie inside a Wasserstein ball around the nominal transition kernel.

# B   Appendix: Lemmas for Neural Tangent Kernel Regime

## B.1   Assumption for NTK Regime

**Assumption 1** (NTK parameterization and initialization). *Each weight matrix $W^{(\ell)} \in \mathbb{R}^{w \times w}$ for $\ell = 1, \ldots, L - 1$ is initialized i.i.d. as $W_{ij}^{(\ell)} \sim \mathcal{N}(0, 1/w)$, and the output layer weights $W^{(L)} \in \mathbb{R}^{1 \times w}$ are initialized i.i.d. as $W_{1j}^{(L)} \sim \mathcal{N}(0, 1/w)$. Biases are initialized i.i.d. $\mathcal{N}(0, 1)$ and are not scaled by $w$.*

This initialization follows the standard Neural Tangent Kernel (NTK) regime, which ensures that the neural network's Jacobian remains nearly constant during training (Jacot et al., 2018). The $1/\sqrt{w}$ scaling of the weights preserves the variance of activations across layers and prevents the vanishing or exploding gradient problems as the network width $w \to \infty$. In the context of our convergence proof, this regime allows the highly non-linear Actor-Critic architecture to be locally approximated by a linear model (Jacot et al., 2018).

**Assumption 2** (Smooth activation). *The activation $\phi : \mathbb{R} \to \mathbb{R}$ is twice continuously differentiable and satisfies* $\|\phi\|_\infty \leq B_0, \quad \|\phi'\|_\infty \leq B_1, \quad \|\phi''\|_\infty \leq B_2$ *for finite constants $B_0, B_1, B_2$.*

This assumption ensures the regularity of the neural network's functional surface. By bounding the first and second derivatives of the activation function, we enforce Lipschitz continuity of both the network output and its gradient with respect to the parameters. In the stochastic approximation framework, these bounds are critical for controlling the variance of the gradient estimators and ensuring that the Hessian of the objective function remains bounded.

**Assumption 3** (Finite input set). *All statements are required only for a finite set $\mathcal{X}$ of inputs with $|\mathcal{X}| < \infty$.*

This assumption is a standard technical requirement in the analysis of Markov Decision Processes (MDPs) to ensure the tractability of the state-action space (Sutton et al., 1998). In the context of our convergence proof, a finite input set $\mathcal{X}$ guarantees that the data distribution possesses a strictly positive lower bound on the probability of visiting each state, provided the underlying Markov chain is ergodic.

**Assumption 4** (Local Parameter Neighborhood in NTK Regime). *Let $\bar{\theta}$ denote the random initialization drawn according to Assumption 1. There exists a universal constant $R > 0$, independent of the network width $w$, such that the network parameters $\theta$ remain within a local Euclidean ball around the initialization:*

$$\mathcal{B}(\bar{\theta}, Rw^{-1/2}) := \left\{ \theta : \|\theta - \bar{\theta}\|_2 \leq Rw^{-1/2} \right\}.$$

*All subsequent theoretical analyses hold for parameter configurations $\theta \in \mathcal{B}(\bar{\theta}, Rw^{-1/2})$.*

## B.2 NTK Linearization of Q function

*Proof of Lemma 1.* For any fixed input $x \in \mathcal{X}$, consider the parameterized path $\theta(\tau) = \bar{\theta} + \tau(\theta - \bar{\theta})$ for $\tau \in [0, 1]$. By the second-order Taylor expansion of $Q_\theta(x)$ around the initialization $\bar{\theta}$, there exists an intermediate point $\tau_x \in (0, 1)$ such that the residual $\mathcal{R}(\theta) = Q_\theta(x) - Q_{\bar{\theta}}(x) - \nabla_\theta Q_{\bar{\theta}}(x)^\top (\theta - \bar{\theta})$ can be expressed in the exact remainder form:

$$\mathcal{R}(\theta) = \frac{1}{2}(\theta - \bar{\theta})^\top \nabla_\theta^2 Q_{\theta(\tau_x)}(x)(\theta - \bar{\theta}).$$

Let $H_{\theta(\tau_x)} := \nabla_\theta^2 Q_{\theta(\tau_x)}(x) \in \mathbb{R}^{d \times d}$ denote the local Hessian matrix. By the matrix-vector inequality $|u^\top M u| \leq \|M\|_\infty \|u\|_1^2$, we bound the absolute residual as $|\mathcal{R}(\theta)| \leq \frac{1}{2}\|H_{\theta(\tau_x)}\|_\infty \|\theta - \bar{\theta}\|_1^2$. Recall that under Assumption 2, the $\ell_\infty$-norm of the Hessian is uniformly stable in the over-parameterized regime. Specifically, as shown in Lee et al. (2019), for any $\theta \in \mathcal{B}(\bar{\theta}, r_1 w^{-1/2})$, there exists a curvature constant $C_H(r_1)$ independent of the width $w$ such that $\sup_{x \in \mathcal{X}, \theta \in \mathcal{B}} \|\nabla_\theta^2 Q_\theta(x)\|_\infty \leq C_H(r_1)w^{-1/2}$ holds with high probability $1 - C_1 \exp(-c_1 w)$.

Since the total number of trainable parameters $d$ satisfies $d = \Theta(w)$, the equivalence of $\ell_p$ norms in $\mathbb{R}^d$ implies $\|\theta - \bar{\theta}\|_1^2 \leq d\|\theta - \bar{\theta}\|_2^2 \leq \alpha w\|\theta - \bar{\theta}\|_2^2$ for some positive constant $\alpha$. Substituting these bounds into the remainder expression yields:

$$|\mathcal{R}(\theta)| \leq \frac{C_H(r_1)}{2\sqrt{w}} \cdot \alpha w \cdot \|\theta - \bar{\theta}\|_2^2.$$

Given the locality constraint $\|\theta - \bar{\theta}\|_2 \leq r_1 w^{-1/2}$, it follows that $\|\theta - \bar{\theta}\|_2^2 \leq r_1^2 w^{-1}$. Thus, the residual simplifies to:

$$|\mathcal{R}(\theta)| \leq \frac{\alpha r_1^2 C_H(r_1)}{2} \cdot \frac{w}{w^{1/2} \cdot w} = \frac{\alpha r_1^2 C_H(r_1)}{2} w^{-1/2} = \mathcal{O}(w^{-1/2}).$$

Finally, since $\mathcal{X}$ is a finite set by Assumption 3, the bound holds uniformly over all $x \in \mathcal{X}$ via a union bound, which completes the proof. $\square$

### B.3 Linearization of the Soft Bellman Error

Recall the soft Bellman error

$$\delta(s, a, s'; \theta) = Q_\theta(s, a) - R(s, a) - \gamma V_\theta(s'),$$

where the soft value function is defined as

$$V_\theta(s') := \log\Big(\sum_{a'} \exp\big(Q_\theta(s', a')\big)\Big).$$

The corresponding softmax policy is given by

$$\pi_\theta(a' \mid s') = \frac{\exp\big(Q_\theta(s', a')\big)}{\sum_{\tilde{a}} \exp\big(Q_\theta(s', \tilde{a})\big)}.$$

By direct differentiation of the log-sum-exp operator, the gradient of the soft value function admits the representation

$$\nabla_\theta V_\theta(s') = \sum_{a'} \pi_\theta(a' \mid s') \, \nabla_\theta Q_\theta(s', a').$$

We state and prove below the linearization of the soft Bellman error used in the main text.

*Proof of Lemma 2.* Recall that

$$\delta(s, a, s'; \theta) = Q_\theta(s, a) - R(s, a) - \gamma V_\theta(s'), \qquad V_\theta(s') = \log\Big(\sum_{a' \in \mathcal{A}} \exp\big(Q_\theta(s', a')\big)\Big),$$

where the action set $\mathcal{A}$ is finite.

By Lemma 1, for any $(s, a)$,

$$Q_\theta(s, a) = Q_{\bar{\theta}}(s, a) + \nabla_\theta Q_{\bar{\theta}}(s, a)^\top (\theta - \bar{\theta}) + r_{s,a}(\theta),$$

where the remainder $r_{s,a}(\theta) = \mathcal{O}(w^{-1/2})$ uniformly over $(s, a)$. Applying the same lemma coordinatewise to $\{Q_\theta(s', a')\}_{a' \in \mathcal{A}}$ yields

$$Q_\theta(s', a') = Q_{\bar{\theta}}(s', a') + \nabla_\theta Q_{\bar{\theta}}(s', a')^\top (\theta - \bar{\theta}) + r_{s',a'}(\theta),$$

with $r_{s',a'}(\theta) = \mathcal{O}(w^{-1/2})$ uniformly over $a' \in \mathcal{A}$.

Define the logit vector

$$q_\theta(s') := \big(Q_\theta(s', a')\big)_{a' \in \mathcal{A}}.$$

By definition,

$$V_\theta(s') = \log\Big(\sum_{a' \in \mathcal{A}} \exp\big(q_{\theta,a'}(s')\big)\Big).$$

The mapping

$$q \mapsto \log\Big(\sum_{a' \in \mathcal{A}} e^{q_{a'}}\Big)$$

is twice continuously differentiable with uniformly bounded Hessian. Therefore, a first-order Taylor expansion at $q_{\bar{\theta}}(s')$ gives

$$V_\theta(s') = V_{\bar{\theta}}(s') + \sum_{a' \in \mathcal{A}} \frac{\exp\big(Q_{\bar{\theta}}(s', a')\big)}{\sum_{\tilde{a} \in \mathcal{A}} \exp\big(Q_{\bar{\theta}}(s', \tilde{a})\big)} \big(Q_\theta(s', a') - Q_{\bar{\theta}}(s', a')\big) + \mathcal{O}(w^{-1/2}).$$

Substituting the linearization of $Q_\theta(s', a')$ into the above expression yields

$$V_\theta(s') = V_{\bar{\theta}}(s') + \Big(\sum_{a' \in \mathcal{A}} \pi_{\bar{\theta}}(a'|s') \, \nabla_\theta Q_{\bar{\theta}}(s', a')\Big)^\top (\theta - \bar{\theta}) + \mathcal{O}(w^{-1/2}),$$

where

$$\pi_{\bar{\theta}}(a'|s') = \frac{\exp\big(Q_{\bar{\theta}}(s', a')\big)}{\sum_{\tilde{a} \in \mathcal{A}} \exp\big(Q_{\bar{\theta}}(s', \tilde{a})\big)}.$$

Substituting the above expansions into the definition of $\delta(s, a, s'; \theta)$ and collecting first-order terms yields

$$\delta(s, a, s'; \theta) = \delta(s, a, s'; \bar{\theta}) + \Big(\nabla_\theta Q_{\bar{\theta}}(s, a) - \gamma \nabla_\theta V_{\bar{\theta}}(s')\Big)^\top (\theta - \bar{\theta}) + \mathcal{O}(w^{-1/2}),$$

which establishes the claimed linearization with

$$\Psi(s, a, s') = \nabla_\theta Q_{\bar{\theta}}(s, a) - \gamma \nabla_\theta V_{\bar{\theta}}(s').$$

$\square$

# C Appendix: Lemmas for Almost Sure Convergence of Network Parameters

In this section, we provide the complete formal proof of Theorem 2.

## C.1 Preliminaries and Notations

Consider a multi-layer neural network Actor-Critic architecture with width $m$. We define the following notations:

- $W_t \in \mathbb{R}^{d_1 \times d_2}$: The weight matrix of the neural network at time step $t$.

- $k_t \in \mathbb{Z}^+$: The rank selected by the adaptive Singular Value Decomposition (SVD) truncation mechanism at step $t$.

- $\beta \in (0, 1)$: The singular value energy threshold used to determine the truncation rank.

- $\Pi_k(\cdot)$: The projection operator onto the non-convex manifold $\mathcal{M}_k$ of rank-$k$ matrices.

- $f(x; W)$: The neural network output function parameterized by $W$ (i.e., the policy output of the Actor or the value output of the Critic). We assume that $f$ is locally $L$-Lipschitz continuous with respect to $W$ within a bounded parameter domain.

- $J(W)$: The objective function to be optimized by the Actor network.

## C.2 Assumption

We now make Lipschitz regularity assumption on the features on the NTK kernel, which is a common assumption in NTK analysis Chen et al. (2020):

**Assumption 5** (Lipschitz regularity). *There exist constants $L, L_b > 0$ such that for all $\mathcal{P}, \mathcal{P}' \in \mathcal{B}(\hat{\mathcal{P}}, \varepsilon)$,*

$$\|A_\mathcal{P} - A_{\mathcal{P}'}\|_2 \leq L\, W^1(\mathcal{P}, \mathcal{P}'), \qquad \|b_\mathcal{P} - b_{\mathcal{P}'}\|_2 \leq L_b\, W^1(\mathcal{P}, \mathcal{P}').$$

**Assumption 6** (Local strong convexity implies quadratic growth). *Let $Q : \mathbb{R}^{d_1 \times d_2} \to \mathbb{R}$ be differentiable and let $W^\star$ be a (local) minimizer. Assume there exist constants $r > 0$ and $\mu > 0$ such that $Q$ is $\mu$-strongly convex on the Frobenius ball $\mathbb{B}_F(W^\star, r) \triangleq \{W : \|W - W^\star\|_F \leq r\}$, i.e., for all $W, W' \in \mathbb{B}_F(W^\star, r)$,*

$$Q(W') \geq Q(W) + \langle \nabla Q(W), W' - W \rangle + \frac{\mu}{2}\|W' - W\|_F^2. \tag{18}$$

*Consequently, for all $W \in \mathbb{B}_F(W^\star, r)$,*

$$\Delta Q(W) \triangleq Q(W) - Q(W^\star) \geq \frac{\mu}{2}\|W - W^\star\|_F^2. \tag{19}$$

### C.3 Lemmas and Proof

#### C.3.1 Proof of Theorem 1

*Proof.* Let $\epsilon_A := A_{\mathcal{P}\circ} - A_{\mathcal{P}}$ and $d := \mathrm{Rank}(A_{\mathcal{P}})$. It follows that:

$$
\begin{aligned}
\theta_r - \theta^\circ &= A_{\mathcal{P},r}^\dagger b_{\mathcal{P}} - \theta^\circ \\
&= A_{\mathcal{P},r}^\dagger (b_{\mathcal{P}} - b_{\mathcal{P}\circ}) + A_{\mathcal{P},r}^\dagger A_{\mathcal{P}\circ} \theta^\circ - \theta^\circ \\
&= A_{\mathcal{P},r}^\dagger (b_{\mathcal{P}} - b_{\mathcal{P}\circ}) + A_{\mathcal{P},r}^\dagger A_{\mathcal{P}} \theta^\circ + A_{\mathcal{P},r}^\dagger \epsilon_A \theta^\circ - \theta^\circ \\
&= A_{\mathcal{P},r}^\dagger (b_{\mathcal{P}} - b_{\mathcal{P}\circ}) + \sum_{i=1}^{r} v_i v_i^T \theta^\circ - \sum_{i=1}^{d} v_i v_i^T \theta^\circ + A_{\mathcal{P},r}^\dagger \epsilon_A \theta^\circ \\
&= A_{\mathcal{P},r}^\dagger (b_{\mathcal{P}} - b_{\mathcal{P}\circ}) - \sum_{i=r+1}^{d} v_i v_i^T \theta^\circ + A_{\mathcal{P},r}^\dagger \epsilon_A \theta^\circ
\end{aligned}
\tag{20}
$$

where the fourth equation above follows from the fact that

$$
A_{\mathcal{P},r}^\dagger A_{\mathcal{P}} = V \Sigma_{\mathcal{P},r}^{-1} U^\top U \Sigma_{\mathcal{P}} V^\top = \sum_{i=1}^{r} v_i v_i^T \quad \text{and} \quad \sum_{i=1}^{d} v_i v_i^T = I
$$

Since $A_{\mathcal{P}}$ is Lipschitz (with constant $L > 0$) in $\mathcal{P} \in \mathcal{B}_W(\hat{\mathcal{P}}, \epsilon)$ we have

$$
\|\epsilon_A\|_2 = \|A_{\mathcal{P}} - A_{\mathcal{P}\circ}\|_2 \le \|A_{\mathcal{P}} - A_{\hat{\mathcal{P}}}\|_2 + \|A_{\hat{\mathcal{P}}} - A_{\mathcal{P}\circ}\|_2 \le 2L\varepsilon
$$

for some $L > 0$. It follows from Eq. (20) that:

$$
\|\theta_r - \theta^\circ\|_2 \;\le\; \|A_{\mathcal{P},r}^\dagger (b_{\mathcal{P}} - b_{\mathcal{P}\circ})\|_2 + \Big\| \sum_{i=r+1}^{d} v_i v_i^T \theta^\circ \Big\|_2 + \|A_{\mathcal{P},r}^\dagger \epsilon_A \theta^\circ\|_2.
$$

Using

$$
\|A_{\mathcal{P},r}^\dagger\|_2 \le \frac{1}{\sigma_{\mathcal{P},r}}, \qquad \|\epsilon_A\|_2 \le 2L\varepsilon,
$$

we get the bound

$$
\|\theta_r - \theta^\circ\|_2 \;\le\; \frac{1}{\sigma_{\mathcal{P},r}} \|b_{\mathcal{P}} - b_{\mathcal{P}\circ}\|_2 + \Big\| \sum_{i=r+1}^{d} v_i v_i^T \theta^\circ \Big\|_2 + \frac{2L\varepsilon}{\sigma_{\mathcal{P},r}} \|\theta^\circ\|_2
$$

Since $b_{\mathcal{P}}$ is Lipschitz in $\mathcal{P}$ with constant $L_b$, then

$$
\|b_{\mathcal{P}} - b_{\mathcal{P}\circ}\|_2 \le 2L_b\varepsilon,
$$

because $\mathcal{P}, \mathcal{P}^\circ \in \mathcal{B}(\hat{\mathcal{P}}, \varepsilon)$. The rank-$r$ truncation suppresses high-variance, low-signal directions of the Bellman operator, improving stability under epistemic uncertainty in dynamics, while introducing bias due to discarded spectral components. This reveals an explicit bias–variance tradeoff governed by the singular value spectrum of $A_{\mathcal{P}}$ $\qquad\square$

#### C.3.2 Proof of Lemmas

*Proof of Lemma 3.* Let $\sigma_1(W_t) \ge \sigma_2(W_t) \ge \cdots \ge \sigma_d(W_t) \ge 0$ denote the singular values of $W_t$. At step $t$, the adaptive energy rule selects

$$
k_{t+1} = \max \left\{ r \in \mathbb{Z}^+ : \frac{\sum_{i=1}^{r} \sigma_i(W_t)}{\sum_{j=1}^{d} \sigma_j(W_t)} \le \beta \right\}.
\tag{21}
$$

By construction, the algorithm enforces a non-inflating rank schedule, i.e.,

$$
k_{t+1} \le k_t, \qquad \forall\, t \ge 0,
\tag{22}
$$

and clearly $k_t \in \mathbb{Z}^+$ for all $t$. Moreover, since $\beta \in (0,1)$ and $W_t \neq 0$ almost surely during training, we have $k_{t+1} \geq 1$, hence

$$k_t \geq 1, \qquad \forall\, t \geq 0. \tag{23}$$

Define

$$k^\star := \inf_{t \geq 0} k_t. \tag{24}$$

Because $\{k_t\} \subset \mathbb{Z}^+$ and Eq. (23) holds, the infimum is well-defined and satisfies $k^\star \in \mathbb{Z}^+$. Since $\{k_t\}$ is nonincreasing by Eq. (22), there exists $T_0 \in \mathbb{N}$ such that $k_{T_0} = k^\star$. Indeed, if $k_t > k^\star$ for all $t$, then $k_t \geq k^\star + 1$ for all $t$ (integrality), implying $\inf_{t \geq 0} k_t \geq k^\star + 1$, a contradiction to Eq. (24). Finally, for any $t \geq T_0$, monotonicity yields

$$k^\star = k_{T_0} \geq k_t \geq \inf_{u \geq 0} k_u = k^\star,$$

so $k_t = k^\star$ for all $t \geq T_0$. This proves that the truncation rank stabilizes in finite time. $\qquad\square$

*Proof of Lemma 4.* For $t \geq T_0$, Lemma 3 establishes that the inherent dimensionality of the network parameters has stabilized at $k^*$. Consequently, any given weight matrix $W \in \mathbb{R}^{d_1 \times d_2}$ in this stabilized phase possesses an exact rank of at most $k^*$.

Let the Singular Value Decomposition (SVD) of $W$ be denoted as $W = U\Sigma V^\top$, where the singular values are sorted in descending order: $\sigma_1 \geq \sigma_2 \geq \cdots \geq \sigma_d \geq 0$, with $d = \min(d_1, d_2)$. Because $\mathrm{rank}(W) \leq k^*$, all singular values beyond the $k^*$-th index are strictly zero:

$$\sigma_i(W) = 0, \quad \forall i \in \{k^* + 1, \ldots, d\} \tag{25}$$

The projection operator $\Pi_{k^*}(\cdot)$ maps the matrix onto the closest rank-$k^*$ subspace. According to the fundamental Eckart-Young-Mirsky Theorem, this projection provides the optimal low-rank approximation in the Frobenius norm, and the squared approximation error is precisely the sum of the squared discarded singular values:

$$\|W - \Pi_{k^*}(W)\|_F^2 = \sum_{i=k^*+1}^{d} \sigma_i^2(W) \tag{26}$$

Substituting the exact singular value condition from Equation (1) into Equation (2), all terms in the summation evaluate to zero:

$$\|W - \Pi_{k^*}(W)\|_F^2 = \sum_{i=k^*+1}^{d} 0^2 = 0 \tag{27}$$

Taking the square root trivially yields $\|W - \Pi_{k^*}(W)\|_F = 0$.

This deterministic result proves that once the adaptive rank algorithm converges to its stable dimension $k^*$, the SVD projection operator acts identically as an identity mapping on the stabilized manifold. Therefore, the projection inherently introduces absolutely zero structural perturbation to the parameters, ensuring pristine gradient flow for the Actor-Critic updates. $\qquad\square$

*Proof of Lemma 5.* For $t \geq T_0$, Lemma 3 guarantees that the adaptive rank of the network has stabilized at the constant $k^*$. Consequently, the subsequent optimization process is mathematically equivalent to executing a Neural Actor-Critic algorithm on a multi-layer neural network with a fixed structure.

By Theorem 3.1 from Tian et al. (2023). Under the standard assumptions of bounded parameter projection and diminishing step sizes (e.g., $\alpha_t = \mathcal{O}(t^{-0.5})$), the expected suboptimality gap of the multi-layer Neural Actor-Critic is asymptotically bounded. Specifically, let $\Delta Q$ denote the expected approximation error of the Q-function. In the general Markov sampling case, as the iteration step $T \to \infty$, the asymptotic upper bound is determined jointly by the network width $m$ and the Critic's irreducible approximation error $\epsilon$:

$$\limsup_{T \to \infty} \mathbb{E}[\Delta Q(W_T)] \leq \lim_{T \to \infty} \left[ \mathcal{O}\left( \frac{(\log T)^2}{\sqrt{T}} \right) \right] + \mathcal{O}(\epsilon) + \tilde{\mathcal{O}}\left( \frac{1}{\sqrt{m}} \right) = \mathcal{O}(\epsilon) + \tilde{\mathcal{O}}\left( \frac{1}{\sqrt{m}} \right) := \epsilon_{gap} \tag{28}$$

Eq. (28) formally implies that the objective sequence converges to an approximate optimal set $\mathcal{W}_\epsilon = \{W \mid \mathbb{E}[\Delta Q(W)] \leq \epsilon_{gap}\}$.

Let $W^* = \arg\min_W \Delta Q(W)$ denote the stationary parameter configuration. We assume that the action-value error function satisfies a local quadratic growth condition with constant $\mu > 0$ within the bounded projection domain:

$$\Delta Q(W) \geq \frac{\mu}{2}\|W - W^*\|_F^2 \tag{29}$$

Applying Jensen's inequality to the expected parameter distance, we establish the explicit mapping from the Q-function error space to the parameter space:

$$\mathbb{E}\left[\|W_t - W^*\|_F\right] \leq \mathbb{E}\left[\sqrt{\frac{2}{\mu}\Delta Q(W_t)}\right] \leq \sqrt{\frac{2}{\mu}\mathbb{E}[\Delta Q(W_t)]} \tag{30}$$

Taking the limit superior on both sides as $t \to \infty$ and substituting the bound from Equation (1), we obtain:

$$\limsup_{t \to \infty} \mathbb{E}\left[\|W_t - W^*\|_F\right] \leq \sqrt{\frac{2}{\mu}\epsilon_{gap}} := \epsilon_W \tag{31}$$

By the $\epsilon$-definition of the limit superior, for any arbitrarily small constant $\eta > 0$, there exists an integer $T_1 \geq T_0$ such that for all $t \geq T_1$:

$$\mathbb{E}\left[\|W_t - W^*\|_F\right] \leq \epsilon_W + \eta \tag{32}$$

Consequently, the expected weight trajectory is strictly confined to an $\epsilon_W$-neighborhood of $W^*$ for all $t \geq T_1$. $\qquad\square$

# D    Appendix: Experimental Results

## D.1    Basic Settings

In all experiments, we evaluate the performance of benchmark algorithms on the `Hopper-v3`, `Walker2d-v3`, `Humanoid-v3`, and `Ant-v3` environments from OpenAI Gym. To ensure a fair comparison, we use the open-source implementation[1] of SAC as the base RL algorithm for all methods, and for RNAC we adopt its original PPO-based trainer without modification. We use Adam as the optimizer in SAC, where both the policy and Q-networks are implemented as two-layer MLPs with hidden sizes $(64, 64)$ and ReLU activation functions. The learning rate for both networks is fixed at $3 \times 10^{-3}$. For our proposed algorithm, we set the truncation interval $d_t$ to $0.7 \times 10^6$ for Walker2d and $10^6$ for Hopper, Humanoid, and Ant, meaning the model is truncated every $d_t$ policy optimization steps. This choice ensures that rank adaptation occurs much less frequently than policy updates.

We apply a standard cosine decay schedule with warm-up, as in (Lialin et al., 2023; Touvron et al., 2023). Specifically, upon each reset, we set the learning rate to zero, gradually warm it up to the target value over 2000 steps, and then resume following the cosine schedule.

We present the practical implementation of our proposed algorithm in Alg. 1. At each iteration, we warm-start both the policy network and Q-network in SAC using the trained neural networks from the previous iteration, and then run SAC in the corresponding MuJoCo environment to continue training.

For the robust RL baselines, we use their official open-source implementations. The implementation of RNAC is available at `https://github.com/tliu1997/RNAC`. To modify the dynamics kernel, we follow the setting in OMPO Luo et al. (2024), using their codebase at `https://github.com/Roythuly/OMPO`. For Parseval regularization, we use the implementation provided at `https://github.com/wechu/parseval_reg`. In MuJoCo experiments with Parseval regularization, we adopt the same setup, tuning the regularization coefficient from $\{0.001, 0.0001, 0.00001\}$ and selecting the best-performing value. We also follow the original implementation by setting $s = 2$ in the Parseval constraint $\|WW^\top - sI\|_F$. For Tiwari et al. (2025), we follow their default configuration with a sparsification layer and set the hidden layer size to 1024 neurons, consistent with their original setting.

---

[1]`https://github.com/openai/spinningup`

## D.2 Rank Constrain for Models

we adopt a low-rank factorization approach (Xu et al., 2019; Zhang et al., 2015): we first factorize it as $W = W^1 W^2$ and then apply singular value decomposition (SVD) to the product $W^1 W^2 = U\Sigma V^\top$. We then reparameterize as $W^1 = U_{[:,:\hat{r}]}\sqrt{\Sigma_{[:\hat{r}]}}$, $W^2 = \sqrt{\Sigma_{[:\hat{r}]}}V_{[:\hat{r},:]}$, where $\hat{r} \leq r$ is the target rank. This projects $W$ onto a lower-rank manifold, thereby enforcing the constraint. As shown in Figure 5, inserting an intermediate linear layer (yellow, within the red region) provides an explicit implementation of this rank reduction. After obtaining the optimized policy $\pi_{W^k}$ from several policy optimization steps, we refine this low-rank representation by performing SVD.

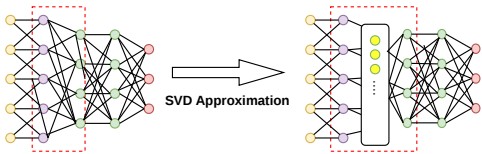

Figure 5: To implement a low-rank constraint, we insert an intermediate linear layer (without activation functions or bias) between the original two layers. This layer acts as a bottleneck that enforces a low-rank factorization of the weight matrix via SVD approximation.

## D.3 Model Uncertainty Setting

Following the setup in Luo et al. (2024), we simulate model uncertainty by introducing continuously varying environment parameters during training. This design encourages policies to generalize across dynamic variations rather than overfitting to a fixed set of dynamics. The specific parameter schedules for each environment are as follows:

- **Hopper:** The torso and foot lengths vary with the episode index $i$ as

$$L_{\text{torso}}(i) = 0.4 + 0.2 \cdot \sin(0.2i), \quad L_{\text{foot}}(i) = 0.39 + 0.2 \cdot \sin(0.2i).$$

- **Walker2d:** The torso and foot lengths follow a similar pattern with

$$L_{\text{torso}}(i) = 0.2 + 0.1 \cdot \sin(0.3i), \quad L_{\text{foot}}(i) = 0.1 + 0.05 \cdot \sin(0.3i).$$

- **Ant:** Gravity $g$ and wind speed $W$ change across episodes according to

$$g(i) = 14.715 + 4.905 \cdot \sin(0.5i), \quad W(i) = 1 + 0.2 \cdot \sin(0.5i).$$

- **Humanoid:** The same variation as Ant is applied, but the wind effect is amplified due to the humanoid's larger mass and drag:

$$g(i) = 14.715 + 4.905 \cdot \sin(0.5i), \quad W(i) = 1 + 0.5 \cdot \sin(0.5i).$$

## D.4 Rank Convergence of the Alternative Algorithm

In this subsection, we conduct an ablation study to examine alternative strategies for selecting the cut-off rank of the SVD beyond Eq. (10). As reviewed by Falini (2022), numerous criteria have been proposed for truncated SVD. Here, we consider a simple hard-thresholding approach based on the ratio between singular values. Specifically, we define the cut-off rank as

$$\hat{r} = \min\left\{ \ell \in \{1, 2, \ldots, d\} \mid \frac{\sigma_\ell}{\sigma_1} \leq \beta \right\}. \tag{33}$$

Figure 7 illustrates a fundamental limitation of this criterion. After the initial iteration, the rank selection process stagnates because the rule in Eq. (33) depends only on the largest singular value. As a result, it ignores the broader spectral structure of the parameters and fails to adapt dynamically to spectral variations during training. Therefore, we continue to use Eq. (10) as our primary rule for selecting the cut-off rank.



Figure 6: Visualization of uncertain dynamics in the Hopper-v3 task, where the torso and foot lengths vary across episodes.

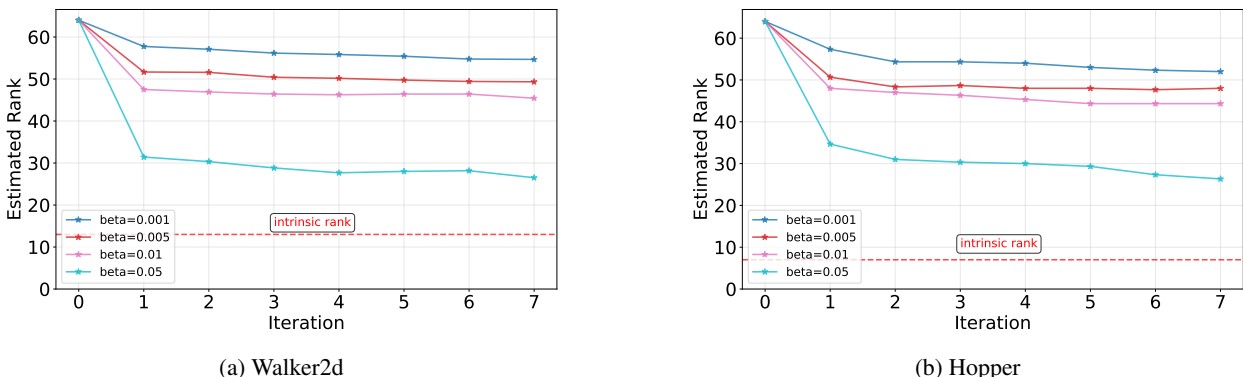

(a) Walker2d

(b) Hopper

Figure 7: Comparison of Rank Selection by hard-thresholding method.

## D.5   Ablation Study of AdaRL

In this subsection, we present additional experimental results under perturbations of physical hyperparameters (e.g., torso length, foot length) in the `Hopper-v3` and `Walker2d-v3` environments.

### D.5.1   Robustness of Policy Models

As shown in Figure 8, the proposed AdaRL algorithm demonstrates consistently superior robustness across a wide range of environments. In particular, it outperforms the strongest baseline (Tiwari et al., 2025) in the majority of settings, achieving higher average returns under varying dynamics.

We observe that AdaRL achieves both higher peak performance and more stable convergence, highlighting its ability to balance expressiveness and robustness via adaptive rank control. Compared to fixed-structure approaches, it better adapts to environmental uncertainty, resulting in more resilient policy learning.

### D.5.2   Impact of Low-Rank Constraints on Different Model Components

We report ablations clarifying where the rank constraints are applied. We evaluate three variants of AdaRL: (i) **actor-only**, where only the policy network is re-factorized; (ii) **critic-only**, where only the value network is re-factorized; and (iii) **both**, which corresponds to the full AdaRL method. For each variant, we apply the factorization described in Figure. 5 to the first two layers while keeping all other components identical.

The results in Table 2 show that applying rank adaptation to either the actor alone or both the actor and critic yields substantial robustness improvements across uncertainty levels and environments. In contrast, the critic-only variant consistently provides limited gains and is often the weakest among the three. This pattern suggests that controlling the expressiveness of the policy network is the primary driver of robustness, while jointly adapting both components offers the most reliable and stable performance. Overall, the ablations demonstrate that AdaRL benefits notably from rank adaptation on the actor side, with the actor–critic configuration delivering the strongest results.

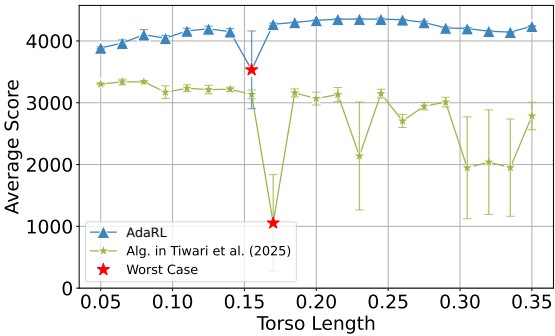

(a) `Walker2d-v3`: Varying torso length with fixed foot length

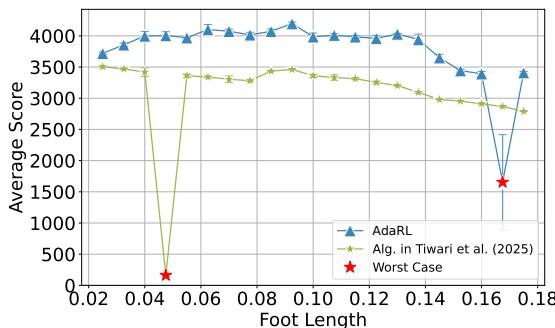

(b) `Walker2d-v3`: Varying foot length with fixed torso length

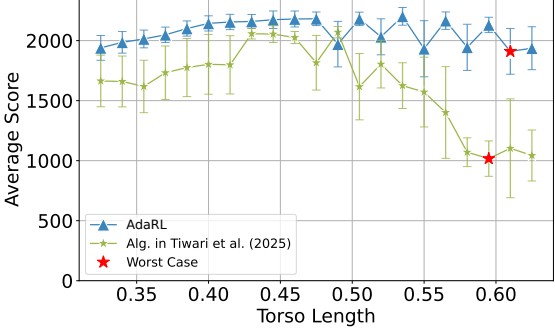

(c) `Hopper-v3`: Varying torso length with fixed foot length

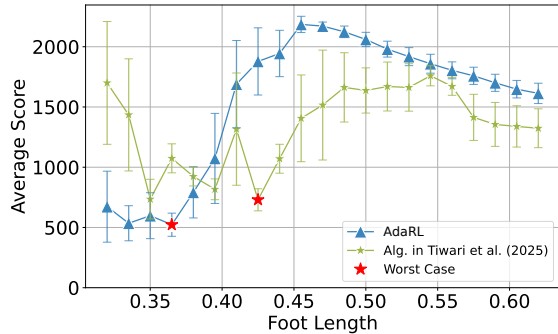

(d) `Hopper-v3`: Varying foot length with fixed torso length

Figure 8: Policy performance under perturbations of physical hyperparameters in `Walker2d-v3` and `Hopper-v3`. Each curve reports the mean performance with shaded regions indicating the standard deviation across seeds. Red pentagram markers (⋆) denote the worst-case performance under each perturbation setting. Subfigures (a) and (c) correspond to varying torso length with fixed foot length, while (b) and (d) show results for varying foot length with fixed torso length. The proposed AdaRL algorithm consistently outperforms the strongest baseline (Tiwari et al., 2025) in most perturbed settings, demonstrating improved robustness.

### D.5.3 Stabilizing AdaRL with a Warm-Start Replay Buffer

To improve the stability of our proposed algorithm and to ensure that the observed performance fluctuations are not caused by changes in environment parameters, we introduce a replay buffer warm-start strategy.

In our initial experiments, we observed a transient performance drop immediately after each rank adaptation step, followed by rapid recovery. To better understand this phenomenon, we found that the replay buffer is effectively reset after each rank update. Consequently, the policy must resample trajectories under newly perturbed dynamics drawn from the Wasserstein uncertainty set. During this early stage, the policy has limited coverage of the underlying transition distribution, causing the SAC algorithm to over-rely on a small set of initial samples. This results in temporary overfitting to a partial subset of the uncertainty set, rather than reflecting a true degradation caused by changes in the environment. As illustrated in the left panel of Figure 9, this effect can be mitigated when a sufficient number of samples are collected.

To address this issue, we adopt a warm-start replay buffer by pre-collecting trajectories under the updated policy before performing policy optimization. As shown in the right panel of Figure 9, this strategy significantly alleviates the transient performance drop and leads to more stable training. Importantly, these results confirm that the previously observed performance fluctuations primarily stem from optimization and data distribution effects, rather than from changes in environment parameters.

Table 2: Performance of the AdaRLvariants across environments and uncertainty levels. Each entry reports the average return over 5 random seeds. Rows highlighted in light blue denote the method achieving the highest performance at Iteration 5.

| Methods | Iteration 1 (0.7e6 Step) | Iteration 3 (2.1e6 Step) | Iteration 5 (3.5e6 Step) |
|---|---|---|---|
| **Hopper-v3 with Low Uncertainty (torso_len $\in [0.31, 0.49]$, foot_len $\in [0.305, 0.485]$)** | | | |
| ALR (Actor Only) | $1308.14 \pm 275.42$ | $2120.85 \pm 115.20$ | $2264.38 \pm 80.59$ |
| ALR (Critic Only) | $1181.78 \pm 197.83$ | $1910.93 \pm 484.25$ | $2076.15 \pm 476.52$ |
| ALR (Actor–Critic Both) | $848.50 \pm 346.82$ | $2205.95 \pm 148.13$ | $2259.59 \pm 40.51$ |
| **Hopper-v3 with High Uncertainty (torso_len $\in [0.25, 0.55]$, foot_len $\in [0.245, 0.545]$)** | | | |
| ALR (Actor Only) | $890.83 \pm 724.68$ | $1614.52 \pm 501.20$ | $1813.32 \pm 223.40$ |
| ALR (Critic Only) | $976.27 \pm 598.22$ | $1778.76 \pm 28.67$ | $1804.34 \pm 255.27$ |
| ALR (Actor–Critic Both) | $608.77 \pm 351.49$ | $1981.51 \pm 214.14$ | $2245.84 \pm 56.32$ |
| **Walker2d-v3 with Low Uncertainty (torso_len $\in [0.1, 0.3]$, foot_len $\in [0.05, 0.15]$)** | | | |
| ALR (Actor Only) | $1780.43 \pm 579.17$ | $3242.00 \pm 247.36$ | $3356.54 \pm 157.38$ |
| ALR (Critic Only) | $1481.17 \pm 692.37$ | $3658.15 \pm 454.64$ | $3699.68 \pm 569.28$ |
| ALR (Actor–Critic Both) | $2171.12 \pm 96.99$ | $4095.72 \pm 83.21$ | $4692.13 \pm 370.46$ |
| **Walker2d-v3 with High Uncertainty (torso_len $\in [0.06, 0.34]$, foot_len $\in [0.03, 0.17]$)** | | | |
| ALR (Actor Only) | $1476.82 \pm 957.33$ | $3644.88 \pm 198.58$ | $3686.93 \pm 160.31$ |
| ALR (Critic Only) | $1253.01 \pm 805.23$ | $3433.09 \pm 189.79$ | $3490.55 \pm 365.96$ |
| ALR (Actor–Critic Both) | $1796.41 \pm 859.72$ | $3146.30 \pm 447.84$ | $3348.37 \pm 270.98$ |

Figure 9: Left: Effect of replay buffer warm-start. Right: Comparison of convergence behavior with and without warm-up initialization.

### D.5.4 Different Activation Function

In this subsection, we evaluate the sensitivity of our proposed algorithm to the choice of activation function. Specifically, we compare two commonly used activations: ReLU and Softplus (Zhou, 2016), where the latter satisfies the differentiability assumptions required in our theoretical analysis. Our results (in Figure 10) indicate that the performance and robustness of the proposed method are largely consistent across both activation functions. This suggests that, despite

the theoretical requirement for smooth (differentiable) activations, the algorithm is not sensitive to the specific choice of activation in practice.

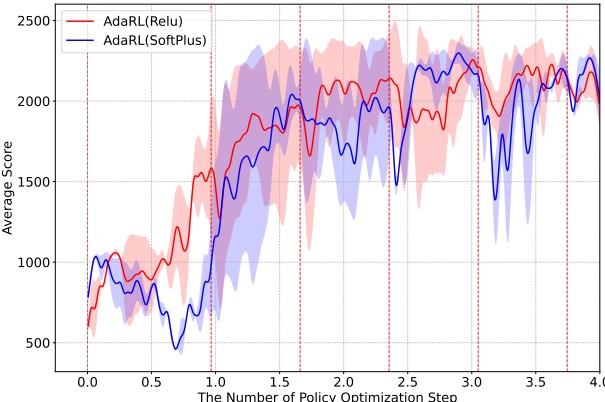

Figure 10: Performance comparison of AdaRL with different activation functions (ReLU vs. SoftPlus). AdaRL with SoftPlus exhibits more stable training dynamics and improved convergence behavior.

