# OpenReview forum: "Adaptive Rank Control for Robust Reinforcement Learning"
_TMLR — Accepted by TMLR_

### Review · Reviewer_ypvE · 2026-03-28

**Summary Of Contributions:**

The authors motivate their work by arguing that classical min-max optimization result in overly conservative policy predictions due to the overcorrection of epistemic uncertainty in the transition dynamics. Consequently, the authors propose an approach that tries to explicitly control for the environment complexity by estimating the local rank of  transition dynamics. The approach leverages a learned proximity metric w.r.t. to the learned parameters and adaptively regularizes training, based on a linearized dynamics rank estimate. Under a set of bounding and smoothness assumptions, the authors are able to bound the (linearized) Bellman error and derive an adaptive variance-bias bound. The authors propose a bi-level optimization, that alternates between estimating the current dynamic's rank for regularizing the model and optimizing the policy. Experiments are carried out over different MuJoCo environments with induced uncertainty/changing environment configurations to simulate epistemic uncertainty. The presented results seem to broadly align with the formerly derived theory and surpass several robust RL baselines.



**Strengths**

1. **Motivation and Approach.** The authors generally argue with the overly conservative nature of existing approaches in terms of predicting actions in uncertain environments. The presented approach of sampling from a set of local dynamics via uniform proximity sampling and combining them into an adaptive rank regularization seems to be sound and to robustly adapt the method's within the different considered environment.
2. **Soundness Proofs.** While reinforcement learning is not my main area of expertise, the proofs for the various lemmas and theorems seem to be well setup and reasonable. Upon checking of the formulas on the high-level, no immediate mistakes in terms of mathematical derivation became obvious to me.
3. **Presentation and Experiments.** Necessary preliminaries on general and entropy-regularized RL are well presented and formalized. I particularly want to highlight the well-setup structure of the paper: starting from initial theory on the shortcomings of 'conservative' min-max approaches, interleaving the theoretical derivation with an experimental validation on the bias-variance trade-off predicted by Thm. 1, finally resulting in the complete approach with experiments and benchmarks.



**Weaknesses**

1. **Potential Limitations**. The authors motivate their method with a possibly overly conservative, worst-case predictions of existing methods and their approach of estimating epistemic uncertainty by sampling within a learned neighborhood. While the authors are generally able to present supporting evidence in terms of overall average performance, robust RL might not only consider average performance, but in many cases be robust to worst-case scenarios. While the authors particularly seem to focus on the former aspect of averaged performance, the paper lacks in consideration of the later aspect. The authors might improve by discussing this trade-off and the algorithm being potentially overly confident.
2. **Evaluation.** In line with the previous point, the authors seem only to test their method under continued training within a novel environment. Given the aforementioned aspect of worst-case robustness, a more fair comparison could be to test model's performance on holdout episodes (if done so already, I would recommend mentioning it in the paper). Considering figure 4, performance for AdaRL seems to steeply drop to (or go below) the level of the SAC on every change of the environment parameterization at at the start of episodes. While the authors seem to attribute these drops to a reset of the optimizer, I would like to see evidence for that. E.g. considering algorithm performance under novel environment without continued training.
3. **Validity of Assumption 4.** Assumption 4 states that 'network parameters $\theta$ [to] remain within a local Euclidean ball around the initialization'. I see no immediate requirement for this assumption and, therefore, consider it rather uncommon or particularly placed to 'make the proof work'. While the assumption and  $Rw^{-1/2}$-bound might hold and be reasonable, the paper lacks justification for it. In addition to this, it is unclear to me during which phase/scenario the parameters $\theta$ are assumed to be constrained: during all of training, for different $\bar{\theta}$ after training, for varying R, all of these scenarios combined, ...? The paper might be improved by better motivating this assumption and, if possible, trying to provide empirical evidence for it to hold within the conducted experiments.
4. **Violation of Assumption 2.** T authors assume a twice continuously differentiable activation function (Assumption 2). However, in their description of experiments (Appendix D.1), they seem to directly violate this assumption by deploying ReLU activations for their MLPs. While one could still assume the theorem to hold approximately, the authors should be more upfront about this and consider the possible impacts of their choice on the results or test with a twice-differentiable activation function in the first place.



**Minor.**

1. The caption of figure 4 does not provide any indication on the environments shown in the two subplots. Assuming, that the plots show results for individual environments, plots for the remaining two environments are missing and should be provided.

**Additional Comments:**

None

**Audience:**

Yes

**Audience Explanation:**

The overall motivation of tackling potential overconservativeness of existing robust RL approaches by estimating and adapting to local dynamic rank provides a novel idea with potential for general performance gains and better stability.

**Broader Impact Concerns:**

Depending on the response to the mentioned weaknesses of algorithm evaluation and robustness, the authors might need to mention the potential instabilities of their approach under changing environments.

**Claims And Evidence:**

No

**Claims Explanation:**

Given the initial motivation of robust RL and considering the presented experiments, I am unsure whether the presented algorithm generally exhibits the robustness properties claimed by the authors. In particular, the potential lack of evaluations under a novel (hold-out) environment parameterizations (as discussed in Weaknesses 1 and 2) in combination with the observed breakdowns on episode boundaries, leave potential questions. I believe this point might be cleared by better specifying evaluation procedure for results in table 1, and (if not already done for the shown results) presenting performance results on novel, hold-out parameterizations.

**Requested Changes:**

My requested changes primarily regard the aforementioned weaknesses:

1. **Potential Limitations.** I would like to recommend further clarifying the indented application scenarios of the approach. How would their algorithm compare to existing methods in terms of worst-case performance?
2. **Evaluation.** Critically, the authors should clarify their procedure of evaluation. The authors should ensure that the observed breakdowns are due to optimizer resets and not due to the changed environment parameters.
3. **Validity of Assumption 4.** Assumption 4 should be better discussed or justified. In particular, the intuition/scenarios why it is assumed to hold is not clear from the present paper.
4. **Violation of Assumption 2.** The authors should state the use of ReLU activations more clearly (leading to a potential mismatch of experiments and theory), discuss the possible impacts on performance and robustness or provide results under differentiable activations.
5. **Link to Proofs (Minor).** The links to the proofs mentioned in Sec. 3.2.1 ("[...] all lemmas are provided in Appendix 1 and 2.") currently link to the lemmas themselves and not to the proofs in the appendix.

---

> ### Author Response · Authors · 2026-04-09
>
> We appreciate your time and effort in reviewing our work. Thank you for your affirmation and constructive feedback.
>
> ### Weaknesses
>
> > 1. Potential Limitations. The authors motivate their method with possibly overly conservative, worst-case predictions of existing methods and estimate epistemic uncertainty by sampling within a learned neighborhood. While supporting evidence is presented for average performance, robust RL may not consider average performance but rather worst-case scenarios. The paper focuses more on averaged performance and lacks discussion of this trade-off. The algorithm may be overly confident.
>
> **Response:** We thank the reviewer for this insightful comment. We agree that classical robust reinforcement learning methods are typically formulated as worst-case (min–max) optimization problems, where performance guarantees are derived with respect to adversarial perturbations of the transition dynamics.
>
> Our work takes a complementary perspective by focusing on representation robustness rather than policy-level worst-case optimality. Specifically, instead of optimizing a policy against the most adverse transition kernel in a Wasserstein uncertainty set, we study how uncertainty in the dynamics affects the statistical estimation of value functions, and how this can be mitigated through low-rank structure.
>
> In the NTK regime, our method can be interpreted as controlling the variance of the soft Bellman error estimator under a distribution of transition kernels. The adaptive rank mechanism suppresses directions in feature space that are weakly supported or highly sensitive to perturbations, thereby improving stability and generalization across the uncertainty set.
>
> This differs fundamentally from min–max approaches, which may be overly conservative by optimizing against worst-case dynamics that are rarely encountered in practice. As a result, our evaluation emphasizes average performance across a distribution of environments, which is consistent with our objective of achieving robust representations under epistemic uncertainty. We agree that this introduces a trade-off: methods optimized for worst-case guarantees may sacrifice average performance, whereas our approach prioritizes statistical efficiency and generalization.
>
> Regarding the concern about overconfidence, we note that the low-rank constraint acts as an implicit regularizer that reduces variance amplification in poorly supported directions of the data distribution. In this sense, the method is designed to be conservative at the representation level, even though it does not explicitly optimize a worst-case control objective.
>
> We will clarify this distinction in the paper.
>
>
> ---
>
> > 2. Evaluation. The method is tested under continued training within a novel environment. A fairer comparison would include holdout episodes. In Figure 4, performance drops are attributed to optimizer reset, but no evidence is provided.
>
> We sincerely thank the reviewer for this insightful comment. Please refer to our response in the “Request to Change” part.
>
> > 3. Validity of Assumption 4. Assumption 4 states that network parameters remain within a local Euclidean ball around the initialization. This is an uncommon and strong assumption, and the paper lacks justification. It is also unclear during which phase this assumption holds.
>
> **Response:** Thanks for your comments. Assumption 4 is consistent with a standard line of analysis in the NTK and lazy-training literature. For example, Theorem 2.1 of Lee et al. (2019) shows that, for sufficiently wide networks, the training dynamics remain well approximated by the linearization around initialization, while the associated Jacobian/NTK quantities stay nearly constant throughout the relevant regime. In this sense, Assumption 4 specifies the local regime in which the first-order analysis is expected to be accurate.
>
> More specifically, the role of Assumption 4 in our paper is to ensure that the iterates remain in a neighborhood of the initialization where higher-order terms are negligible and the local approximation remains valid. This type of near-initialization condition is standard in analyses based on NTK-style linearization, where controlling the parameter drift is essential for preserving kernel stability and justifying the approximation.
>
>
> **Reference:** Lee et al. (2019), *Wide Neural Networks of Any Depth Evolve as Linear Models Under Gradient Descent*, *Advances in Neural Information Processing Systems (NeurIPS)*, 32.

---

> > ### Author Response · Authors · 2026-04-09
> >
> > > 4. Violation of Assumption 2. The paper assumes twice continuously differentiable activation functions, but uses ReLU in experiments, which violates this assumption.
> >
> > **Response:** We thank the reviewer for pointing this out. Assumption 2 is a technical condition used in the theoretical analysis to control higher-order remainder terms, such as those arising in Taylor or curvature-based arguments. While ReLU is not $C^2$ at $0$, it is piecewise linear and differentiable almost everywhere; the nondifferentiable set has measure zero and therefore does not affect the gradient- or NTK-style quantities considered in expectation.
> >
> > Moreover, the theory can be made fully consistent with standard smoothing arguments: replacing ReLU with a $C^\infty$ approximation such as Softplus
> >
> > $$
> > \phi_\tau(x) = \tau \log\bigl(1 + e^{x/\tau}\bigr),
> > $$
> >
> > satisfies Assumption 2, and $\phi_\tau \to \mathrm{ReLU}$ uniformly as $\tau \to 0$, with the resulting bounds varying continuously in $\tau$. Empirically, we use ReLU for comparability with common baselines; the observed trends do not depend on twice continuous differentiability.
> >
> > ---
> >
> > ### Minor Issues
> >
> > > 1. The caption of Figure 4 does not specify the environments shown in the subplots, and results for two environments appear to be missing.
> >
> > **Response:** We thank the reviewer for pointing this out. We have corrected the caption of Figure 4 to clearly specify the environments shown in each subplot and ensured that all relevant results are properly included in the revised version.
> >
> > ---

---

> > > ### Author Response · Authors · 2026-04-09
> > >
> > > > 1. Potential Limitations. I would like to recommend further clarifying the intended application scenarios of the approach. How would their algorithm compare to existing methods in terms of worst-case performance?
> > >
> > > **Response:** We thank the reviewer for this insightful comment.
> > >
> > > Our approach targets continuous control problems with epistemic uncertainty, where explicit worst-case optimization is often intractable [1]. Instead, we adopt a practical alternative by sampling transition dynamics from the uncertainty set and controlling policy complexity via adaptive rank selection.
> > >
> > > While our method does not explicitly optimize worst-case return, it improves robustness by reducing sensitivity to epistemic perturbations. Empirically, as shown in Appendix D.5.1, it also achieves good performance under held-out and perturbed dynamics. We will further clarify this in the revised manuscript.
> > >
> > > [1]Zhou, Ruida, et al. "Natural actor-critic for robust reinforcement learning with function approximation." Advances in neural information processing systems 36 (2023): 97-133.
> > >
> > > ---
> > >
> > > > 2. Evaluation. Critically, the authors should clarify their procedure of evaluation. The authors should ensure that the observed breakdowns are due to optimizer resets and not due to the changed environment parameters.
> > >
> > > **Response:** We thank the reviewer for this important and insightful comment.
> > >
> > > Motivated by this suggestion, we revisited our experiments and conducted additional analysis to better understand the source of the observed performance drops. After careful investigation, we found that the issue is not caused by optimizer resets, but rather stems from a sampling-related effect associated with replay buffer updates.
> > >
> > > Specifically, after each rank adaptation step, the replay buffer is effectively reset, meaning that most previously collected transitions are no longer used. As a result, the agent must quickly collect new trajectories under the updated policy and perturbed dynamics. In the early stage, the buffer contains only a small number of samples, which do not adequately cover the underlying transition distribution. This leads the SAC updates to rely on a limited and potentially biased subset of data, causing temporary overfitting and a short-term drop in performance. As more diverse samples are collected, the data distribution becomes more representative, and the performance recovers accordingly.
> > >
> > > We sincerely appreciate this comment, which helped us identify the true underlying cause and improve both our understanding and the stability of the algorithm. In particular, we introduce a replay buffer warm-start strategy that significantly mitigates this issue. We have added new experimental results in Appendix D.5.3 to support this finding.
> > >
> > > ---
> > >
> > > > 3. Validity of Assumption 4. Assumption 4 should be better discussed or justified. In particular, the intuition/scenarios why it is assumed to hold is not clear from the present paper.
> > >
> > > **Response:** Thank you for this comment. The intuition behind this assumption is that our analysis is intended to describe a local NTK/lazy-training regime, rather than the full nonlinear training trajectory over arbitrary time horizons. In this regime, the parameter updates remain sufficiently small so that the network stays close to its random initialization, the Jacobian/NTK varies little, and the Q-function and soft Bellman error are both well approximated by their first-order expansions.
> > >
> > > This type of assumption is standard in NTK-based analyses. In particular, for sufficiently wide networks, results such as Theorem 2.1 of Lee et al. (2019) show that gradient descent dynamics can remain close to the linearized model around initialization, with the associated kernel remaining nearly constant throughout the relevant phase of training. From this perspective, Assumption 4 is not meant to describe all stages of practical training, but rather to specify the phase in which the local linearization argument is theoretically justified.
> > >
> > > In our paper, Assumption 4 is used precisely to control higher-order terms and preserve the stability of the local kernel geometry needed in Lemma 1 and Lemma 2.
> > >
> > > **Reference:** Lee et al. (2019), *Wide Neural Networks of Any Depth Evolve as Linear Models Under Gradient Descent*, *Advances in Neural Information Processing Systems (NeurIPS)*, 32.

---

> > > > ### Author Response · Authors · 2026-04-09
> > > >
> > > > > 4. Violation of Assumption 2. The authors should state the use of ReLU activations more clearly (leading to a potential mismatch of experiments and theory), discuss the possible impacts on performance and robustness or provide results under differentiable activations.
> > > >
> > > > **Response:** We thank the reviewer for this important comment.
> > > >
> > > > We acknowledge that Assumption 2 requires smooth activation functions, whereas ReLU is used in our empirical implementation. To address this potential mismatch, we have clarified this point in the revised manuscript and conducted additional experiments using smooth activation functions (e.g., Softplus).
> > > >
> > > > The results show that the performance and robustness of our method are largely insensitive to the choice of activation function, with Softplus yielding comparable results to ReLU. This suggests that, while smoothness is required for theoretical analysis (e.g., NTK-based arguments), the practical performance of the proposed algorithm does not critically depend on this assumption.
> > > >
> > > > For consistency and fair comparison, we retain ReLU in the main experiments, as it is also adopted by all baseline methods. Additional results with smooth activations are provided in Appendix D.5.4.
> > > >
> > > > ---
> > > >
> > > > > 5. Link to Proofs (Minor). The links to the proofs mentioned in Sec. 3.2.1 ("[...] all lemmas are provided in Appendix 1 and 2.") currently link to the lemmas themselves and not to the proofs in the appendix.
> > > >
> > > > **Response:** We thank the reviewer for the careful proofreading. We have corrected the links to ensure they point to the corresponding proofs in the appendix.

---

### Review · Reviewer_6Cs6 · 2026-03-30

**Summary Of Contributions:**

This paper proposes a novel framework for robust reinforcement learning that leverages adaptive low-rank adapters. The authors point out a significant "bias-variance trade-off" in uncertain environments that high-rank models typically overfit perturbations while low-rank models typically lack expressive power and introduce bias. To address this, the proposed framework employs a bi-level optimization mechanism. Experimental results demonstrate that the proposed method significantly outperforms existing baselines on the MuJoCo benchmarks.

**Audience:**

Yes

**Audience Explanation:**

The bi-level optimization mechanism is well motivated. Experimental results are quite sufficient, demonstrating that the proposed method significantly outperforms existing baselines.

**Broader Impact Concerns:**

I do not see noticeable broader impact concerns.

**Claims And Evidence:**

Yes

**Claims Explanation:**

1.	The overall presentation is clear and well-structured.
2.	This paper not only provides an intuitive explanation but also rigorously proves the bias-variance trade-off bounds caused by rank selection in RL under epistemic uncertainty.
3.	The experimental results are compelling, with the proposed method outperforming all baselines across MuJoCo tasks.

**Requested Changes:**

1.	A noticeable performance degradation occurs immediately after each rank adaptation step, requiring substantial recovery time before improvement resumes. This knowledge-forgetting effect is acknowledged but not addressed. Could the authors discuss potential solutions to mitigate this issue or accelerate recovery?
2.	Since the authors claim reduced computational complexity, a comparison of computational cost is highly suggested to better understand the efficiency of the proposed method.

---

> ### Author Response · Authors · 2026-04-09
>
> We greatly appreciate the careful proofreading of the manuscript.
>
>
> > 1. A noticeable performance degradation occurs immediately after each rank adaptation step, requiring substantial recovery time before improvement resumes. This knowledge-forgetting effect is acknowledged but not addressed. Could the authors discuss potential solutions to mitigate this issue or accelerate recovery
>
> **Response**: We thank the reviewer for this important and insightful comment.
>
> Motivated by this suggestion, we revisited our experiments and conducted additional analysis to better understand the source of the observed performance drops. After careful investigation, we found that the issue is not caused by optimizer resets, but rather stems from a sampling-related effect associated with replay buffer updates.
>
> Specifically, after each rank adaptation step, the replay buffer is effectively reset, meaning that most previously collected transitions are no longer used. As a result, the agent must quickly collect new trajectories under the updated policy and perturbed dynamics. In the early stage, the buffer contains only a small number of samples, which do not adequately cover the underlying transition distribution. This leads the SAC updates to rely on a limited and potentially biased subset of data, causing temporary overfitting and a short-term drop in performance. As more diverse samples are collected, the data distribution becomes more representative, and the performance recovers accordingly.
>
> We sincerely appreciate this comment, which helped us identify the true underlying cause and improve both our understanding and the stability of the algorithm. In particular, we introduce a replay buffer warm-start strategy that significantly mitigates this issue. We have added new experimental results in Appendix D.5.3 to support this finding.
>
> > 2. Since the authors claim reduced computational complexity, a comparison of computational cost is highly suggested to better understand the efficiency of the proposed method.
>
> **Response**: We thank the reviewer for this important comment.
>
> Our method reduces computational overhead by avoiding per-step SVD operations. Rather than enforcing the low-rank constraint through repeated SVD on parameters or gradients, we impose it directly through the model structure and only update the rank occasionally. In practice, SVD is performed very infrequently (typically fewer than 10 times over the entire training process), so its additional cost is negligible compared to standard training.
>
> To make this clearer, we summarize the computational cost below:
>
> | Method | Computational Cost |
> |--------|------------------|
> | RL baseline | training steps × backward pass |
> | Ours | training steps × backward pass + rank adaptation steps × SVD cost (typically < 10 steps) |
> | per-step SVD | training steps × (backward pass + SVD cost) |
>
> We will include this clarification in the revised manuscript to better highlight the computational advantage of our approach.

---

### Review · Reviewer_4Ynx · 2026-04-01

**Summary Of Contributions:**

The paper provided an interesting alternative perspective on robustness in reinforcement learning, in contrast to classic min-max worst-case analysis. Their framework tries to learn policies under transition dynamics sampled from an epistemic uncertainty set. This natural gives us the question of “how uncertainty in the transition model interacts with the complexity of the policy representation” and “how the rank of the model relates to the tradeoff between robustness and expressiveness”. In particular,  they employ sampling transition dynamics from a Wasserstein uncertainty set, which induces a form of epistemic regularization, rather than worst-case pessimism.  This is implemented using a bi-level optimization framework. At the lower level, we optimize a policy under transition dynamics sampled from a Wasserstein ball, given the fixed low-rank constraint on the policy representation. At the upper level, the learner adapts the rank to balance robustness and expressiveness. This bi-level optimization problem itself is hard to solve exactly. The authors proposed the specific algorithm procedure for Rank Adaptation Step by empirical and greedy search algorithm combined with a low-rank factorization approach.  Finally, the authors demonstrated empirical results on MuJoCo continuous-control benchmarks.  AdaRL is compared  against several robust RL baselines, including RNAC, Parseval regularization, fixed-rank SAC, and the algorithm from Tiwari et al. (2025).

**Audience:**

Yes

**Audience Explanation:**

The paper provided an interesting alternative perspective on robustness in reinforcement learning, in contrast to classic min-max worst-case analysis, by adaptively controlling the policy's representational rank during training.

**Broader Impact Concerns:**

The paper does not raise possible ethical issues.

**Claims And Evidence:**

Yes

**Claims Explanation:**

Strength

- Novelty:  Theorem 1 provides a characterization of a bias-variance trade-off in entropy-regularized RL under epistemic uncertainty when policy models are restricted to be of a low rank:

$$||\theta_r - \theta^\circ||_2 \le \frac{2\epsilon(L_b + L||\theta^\circ||_2)}{\sigma_{\mathcal{P},r}} + \left|\left|\sum_{i=r+1}^d v_i v_i^\top \theta^\circ\right|\right|_2$$

 Section 3.3 Illustration of Theorem 1 is also helpful to demonstrate such the trade-off.

- Theorem 2 guarantees almost sure convergence of network parameters on $\epsilon_W$-neighborhood of the stationary point $W^*$.


- Writing and Clarity: The paper is well-structured and covers the literature adequately. The details on the formulation and algorithm design are easy to follow. The explanation on theoretical analysis is also nice to read. I did not see any notation issues or critical typos in the main paper.

**Requested Changes:**

Weakness
- Theoretical analysis heavily relies on NTK Linearization. Lemma 1 for Q-function and Lemma 2 for Soft Bellman Error are derived via NTK regime and then we can ignore $O(w^{-1/2})$ term. This allows the remaining part just consider the linear least-squares problem. The analysis and algorithm design is very nice. I wonder if the authors see any limitations on using NTK or if you think this is a reasonable modeling.
- Algorithm design relies on stationary environments, as the rank update is not flexible.


Request Changes

Future work and discussion on the limitations would be greatly appreciated to make the paper complete. It might be related to extensions to non-linear model, non-stational setting, scalability.


Questions

How is it possible to sample over the Wasserstein ball? In experiments, do we sample approximately, or do we have a specific structure to conduct it?

---

> ### Author Response · Authors · 2026-04-09
>
> We greatly appreciate the careful proofreading of the manuscript.
>
> > 1. Theoretical analysis heavily relies on NTK Linearization. Lemma 1 for Q-function and Lemma 2 for Soft Bellman Error are derived via NTK regime and then we can ignore \(O(w^{-1/2})\) term. This allows the remaining part just consider the linear least-squares problem. The analysis and algorithm design is very nice. I wonder if the authors see any limitations on using NTK or if you think this is a reasonable modeling.
>
> **Response:** Thank you for this thoughtful comment. We agree that the NTK/lazy-training regime is not the only route to obtaining the bias-variance trade-off.
>
> In particular, one can also consider fine-tuning a pre-trained reference network with parameters $\hat{\theta}=(\theta_1,\hat\theta_{-1})$, where $\theta_1$ corresponds to the parameters of the first linear layer and $\hat\theta_{-1}$ corresponds to all other parameters in the network. A first-order expansion around $\hat{\theta}_1$ is
>
> $$
> Q_{(\theta_1,\hat\theta_{-1})} (s,a) \approx Q_{\hat{\theta}}(s,a)+ \langle \nabla_{\theta_1} Q_{\hat{\theta}}(s,a),\,\theta_1-\hat \theta_1\rangle,
> $$
>
> and the soft Bellman error admits a corresponding linearization
>
> $$
> \varepsilon(s,a,s';\theta_1,\hat\theta_{-1})
> \approx
> \varepsilon(s,a,s';\hat{\theta}_1,\hat\theta_{-1})
> +
> \langle \Psi_{\hat\theta}(s,a,s'),\,\theta_1-\hat \theta_1\rangle,
> $$
>
> where
>
> $$ \Psi_{\hat{\theta}}(s,a,s^{\prime}) = \nabla_{\theta_1} Q_{\hat{\theta}}(s,a) - \gamma \sum_{a^{\prime}} \pi_{\hat{\theta}}(a^{\prime}\mid s^{\prime}) \nabla_{\theta_1} Q_{\hat{\theta}}(s^{\prime},a^{\prime}). $$
>
> After vectorization, this again leads to a local linear least-squares problem with first-order condition
>
> $$
> A^{\mathrm{loc}}_P(\varrho-\hat\varrho)=b^{\mathrm{loc}}_P.
> $$
>
> and thus to the same qualitative low-rank bias--variance tradeoff as in our NTK-based analysis. In this sense, the phenomenon is not unique to NTK, but can also arise when fine-tuning the first layer of a pre-trained reference network.
>
> In future research we will consider the case in which **all** parameters of the reference network are fine-tuned and the first layer is subject to a rank constraint. In such setting, the analysis needs to consider the restricted Hessian of the soft-Bellman loss along the fixed rank manifold. Under local smoothness and nondegeneracy assumptions, the excess error relative to the unconstrained optimum decomposes into a bias term given by the Hessian-weighted projection onto the fixed rank model class and a variance term governed by the inverse restricted Hessian and the size of the transition-model perturbation.
>
>
>
>
> ---
>
> > 2. Algorithm design relies on stationary environments, as the rank update is not flexible.
>
> **Response:** We thank the reviewer for raising this important point.
>
> Our setting does not assume a strictly stationary environment at the training level. Instead, we consider a family of MDPs induced by sampling transition kernels from a Wasserstein uncertainty set. Conditioned on a sampled kernel, each episode is stationary; however, across episodes the dynamics vary, introducing controlled non-stationarity during training. The policy and critic are trained using a replay buffer that aggregates transitions from both current and past episodes. As a result, the effective data distribution is not tied to a single MDP, but rather corresponds to a mixture distribution over transition kernels. In the NTK regime, this induces a quasi-stationary linear regression problem, with a covariance structure that evolves slowly as new data are incorporated.
>
> The adaptive rank mechanism operates on this induced feature covariance. Since the replay buffer smooths distributional changes across episodes, the covariance spectrum does not change abruptly, and the rank updates remain stable. In this sense, the rank adaptation is not required to track instantaneous environment changes, but rather to capture the dominant directions of variability in the aggregated data distribution. Therefore, while the environment is non-stationary across episodes, the learning problem faced by the critic is effectively slowly varying, which is compatible with our rank adaptation scheme. Moreover, the presence of variability across transition kernels is precisely what makes low-rank regularization beneficial, as it controls variance in directions that are weakly supported across the mixture distribution. This setting is consistent with standard robustness evaluation protocols in reinforcement learning, where environment parameters are randomized across episodes (e.g., [1]).
>
> [1]Gu, Shangding, et al. "Robust gymnasium: A unified modular benchmark for robust reinforcement learning." arXiv preprint arXiv:2502.19652 (2025).

---

> > ### Author Response · Authors · 2026-04-09
> >
> > > 3. Future work and discussion on the limitations would be greatly appreciated to make the paper complete. It might be related to extensions to non-linear model, non-stational setting, scalability.
> >
> > **Response:** Thanks for this question. The current paper should be viewed as a first step in a broader research program. We plan to explore the following directions:
> >
> > (1) Beyond first-order analysis.
> > Our current analysis relies on a first-order (NTK) approximation. A natural extension is to incorporate higher-order effects to better capture sensitivity to model perturbations and further understand the role of low-rank constraints in improving robustness.
> >
> > (2) Handling non-stationarity.
> > Another direction is to explicitly account for the non-stationary data distribution induced by replay buffers and changing dynamics. This could enable more principled adaptation of the representation to evolving data distributions.
> >
> > > 4. How is it possible to sample over the Wasserstein ball? In experiments, do we sample approximately, or do we have a specific structure to conduct it?
> >
> > **Response:** Thank you for this important question. In our theoretical development, the Wasserstein ball should be interpreted as a local ambiguity set over transition kernels. In simple settings such as linear dynamical systems (e.g., linearized CartPole), such ambiguity sets can be characterized analytically.
> >
> > However, in more complex environments such as MuJoCo, the transition dynamics are nonlinear and/or not available in closed form, making it infeasible to directly construct or sample from a Wasserstein ball over transition kernels. Instead, we approximate this uncertainty by sampling physical parameters of the simulator (e.g., gravity, link lengths) within a neighborhood of their nominal values. Each sampled parameter realization induces a different transition kernel, generating a family of dynamics around the nominal model. This defines a parametric approximation of a local ambiguity set, which can be viewed as a structured subset of a Wasserstein neighborhood.
> >
> > From the perspective of the training distribution, this procedure induces a mixture distribution over transitions (s,a,s’), which is consistent with the assumptions underlying our analysis. Importantly, the transition kernel remains unknown to the agent, and this parameter randomization provides a practical way to model epistemic uncertainty in simulation.
> >
> > We will clarify this connection between the Wasserstein formulation and its parametric approximation in the revised manuscript.

---

### Review · Reviewer_oCek · 2026-04-01

**Summary Of Contributions:**

The paper addresses RL in environments with epistemic uncertainty and develops a theory to decompose the parameter estimation error into variance and bias contributions in terms of the truncation rank of the parameters under this setup. Based on this theory which highlights that truncation rank affects the bias-variance tradeoff, the authors propose an adaptive rank control algorithm to determine the optimal rank for a specific RL task and optimize for the model parameters under the corresponding rank constraints. The algorithm is formulated as a bi-level search problem and the outer optimization for the rank is performed through a relatively simple greedy search. Experiments are conducted on variants of several OpenAI Gym environments showing the advantage of the proposed methods over standard baselines when epistemic uncertainty is involved in these RL environments.

**Audience:**

Yes

**Audience Explanation:**

Addressing uncertainty in transition dynamics is an important and relevant topic for the audience in robust reinforcement learning.

**Claims And Evidence:**

Yes

**Claims Explanation:**

The paper claims that there is a bias-variance trade-off in the parameter estimation error for RL environments with uncertain transition dynamics, and that the model complexity (in terms of the rank of the parameter matrices) needs to be explicitly controlled to achieve the best performance, as indicated by such a trade-off. This is supported through extensive and well-designed experiments on standard benchmark environments, only modified to introduce the epistemic uncertainty matching the theoretical setup.

**Requested Changes:**

* Sec 3.2.1: It would be helpful to summarize the assumptions 1-4 in the main text (informally). This enables readers to have a rough understanding of the practical limits of the theorems that follow.
* Sec 3.3: Please elaborate on why the model with rank=8 achieving the best performances "closely aligns with theoretical prediction".
* Sec 3.3: One of my questions is how exactly the result in Example 1 demonstrates the bias-variance "tradeoff". There is only a single performance metric, which does not illustrate the two-term decomposition in Thm 1. The same applies to other experiments.
* Another question is whether the modified CartPole system represents the most general scenario with epistemic uncertainty. More specifically, even though the pole length is randomly sampled across episodes, it remains *fixed* within each episode. In principle, you are not sampling the transition kernel $\mathcal P_{s_t, a_t}$ i.i.d. for different timesteps $t$ within a single trajectory. I'm curious whether this would bring up any issue when applying your theory.
* Sec 4.2: The description of the Rank Adaptation Step is somewhat verbose for a simple greedy search strategy.
* Minor formatting issues & typos:
    * Use \eqref instead of \ref for equations
    * Add spacing after paragraph header. Maybe use \paragraph instead of simply \textbf
    * Sec 3.2.1 the end of first paragraph: "1 and 2" looks out of place

---

> ### Author Response · Authors · 2026-04-09
>
> We wish to thank the reviewer for their time and careful reading of our manuscript!
>
> > Sec 3.2.1: It would be helpful to summarize the assumptions 1-4 in the main text (informally). This enables readers to have a rough understanding of the practical limits of the theorems that follow.
>
> **Response:** Thank you for this helpful suggestion. We agree that a summary of Assumptions 1--4 in the main text improves readability. In the revised manuscript, we have added a concise informal discussion of these assumptions directly in Section 3.2.1, including the NTK initialization/scaling, smoothness of the activation, the finite input-set setting, and the local neighborhood condition around initialization.
>
> ---
>
> > Sec 3.3: Please elaborate on why the model with rank=8 achieving the best performances "closely aligns with theoretical prediction".
>
> **Response:**  We thank the reviewer for this comment which allows us to clarify our work.
>
> The statement refers to the bias–variance trade-off characterized in Theorem 1. In the NTK regime, the soft Bellman error can be approximated as a linear regression problem, where the generalization error decomposes into a bias term, which decreases with rank due to increased representational capacity, and a variance term, which increases with rank due to amplification of noise and sensitivity to epistemic uncertainty (particularly along weakly excited directions of the feature covariance).
> As a result, the theory predicts that the optimal performance should occur at an intermediate rank that balances these two effects, rather than at either extreme.
>
> In the CartPole experiment, the nominal linearized system has intrinsic state dimension 4, which suggests that a rank-4 model would be sufficient in the absence of uncertainty. However, the introduction of epistemic uncertainty (via parameter randomization) effectively enriches the distribution of transition dynamics, increasing the dimensionality of the induced feature covariance. In particular, variability across transition kernels introduces additional directions in feature space that are relevant for prediction but are also more weakly supported.
>
> A rank-4 model is therefore too restrictive (high bias), while higher-rank models capture these additional directions but incur increased variance. Empirically, rank = 8 provides the best trade-off: it is large enough to represent the dominant variability induced by the uncertainty, but small enough to avoid variance amplification in poorly supported directions.
>
> In this sense, the observed optimal rank aligns with the theoretical prediction that performance is maximized at an intermediate rank determined by the spectrum of the feature covariance under the induced data distribution.
>
>
> ---
>
> > Sec 3.3: One of my questions is how exactly the result in Example 1 demonstrates the bias-variance "tradeoff". There is only a single performance metric, which does not illustrate the two-term decomposition in Thm 1. The same applies to other experiments.
>
> **Response:** We thank the reviewer for this insightful comment.
>
> We agree that directly visualizing the bias–variance decomposition in Theorem 1 is challenging in reinforcement learning. In particular, the optimal parameterization is generally not unique, and the true parameter error (as characterized in the theorem) is not directly observable in practice.
>
> For this reason, we follow a standard approach and use policy performance (i.e., return) as a proxy for policy quality. While this does not allow us to explicitly separate the two terms, it provides an informative empirical signal. Under this view, the bias–variance trade-off is reflected indirectly: low-rank models tend to underfit (high bias), while high-rank models become more sensitive to perturbations in transition dynamics (high variance), leading to degraded performance under uncertainty.
>
> Therefore, the consistent observation that performance peaks at intermediate ranks aligns with the qualitative prediction of Theorem 1. Although we do not directly visualize the two-term decomposition, the empirical trend matches the expected bias–variance trade-off under epistemic uncertainty.
>
> ---

---

> > ### Author Response · Authors · 2026-04-09
> >
> > > Another question is whether the modified CartPole system represents the most general scenario with epistemic uncertainty. More specifically, even though the pole length is randomly sampled across episodes, it remains fixed within each episode. In principle, you are not sampling the transition kernel \(P_{s_t,a_t}\) i.i.d. for different timesteps \(t\) within a single trajectory. I'm curious whether this would bring up any issue when applying your theory.
> >
> > **Response:** The reviewer raises an important point regarding the temporal structure of epistemic uncertainty. In our theoretical analysis, transitions  $(s,a,s’)$ are sampled from the steady-state distribution induced by a behavior policy, with a transition kernel drawn from a Wasserstein ambiguity set around a nominal model. In particular, the analysis does not assume that the transition kernel is resampled independently at each timestep within a trajectory, but rather that the overall sampling distribution over transitions reflects variability in the underlying dynamics.
> >
> > In the modified CartPole environment, the pole length is randomly sampled at the beginning of each episode and remains fixed within that episode. When combined with a replay buffer that aggregates transitions across many episodes and policy iterates, the effective training distribution for the critic becomes a mixture distribution.  From the perspective of the NTK-based linearized soft Bellman error, the critic is trained on samples $(s,a,s’)$ drawn from this mixture distribution. This is consistent with the theoretical setting, where the expectation is taken over a distribution of transition kernels within a Wasserstein neighborhood. While the CartPole parameterization induces a structured (low-dimensional) subset of this ambiguity set, it still provides variability in next-state transitions sufficient to probe the bias–variance tradeoff governed by the induced feature covariance.
> >
> > Therefore, although the transition kernel is fixed within each episode, the combination of cross-episode randomization and replay-buffer sampling yields an effective distribution over transitions that aligns with the assumptions of the theoretical analysis.
> >
> >
> >
> > ---
> >
> > > Sec 4.2: The description of the Rank Adaptation Step is somewhat verbose for a simple greedy search strategy.
> >
> > **Response:** We thank the reviewer for this helpful comment. To improve readability, we have streamlined the main text and moved some of the detailed implementation discussions to the appendix.
> >
> > ---
> >
> > > Minor formatting issues & typos:
> > > - Use \eqref instead of \ref for equations
> > > - Add spacing after paragraph header. Maybe use \paragraph instead of simply \textbf
> > > - Sec 3.2.1 the end of first paragraph: "1 and 2" looks out of place
> >
> > **Response:** We thank the reviewer for highlighting these minor formatting issues and typos. We have addressed all of them in the revised manuscript.

---

### Decision · Action_Editor_axDU · 2026-05-13

**Recommendation:** Accept as is

**Audience:**

Yes

**Audience Explanation:**

The paper studies an important problem in robust reinforcement learning: how to obtain robustness to epistemic transition uncertainty without relying on computationally expensive and potentially overly conservative min--max optimization. The proposed perspective of controlling policy/value-function complexity through adaptive low-rank representations is interesting and relevant to researchers working on robust RL, regularization in RL, uncertainty-aware decision making, and the theory/practice interface of deep RL.

The work is somewhat specialized, and its theory is tied to a local/NTK-style regime. Nevertheless, the paper offers a coherent alternative to adversarial robust RL, and the empirical results suggest that adaptive rank selection can improve robustness and efficiency in benchmark continuous-control settings. This should be of interest to part of the TMLR audience.

**Claims And Evidence:**

Yes

**Claims Explanation:**

The reviewers converged that the paper’s main claims are supported by the theoretical development and empirical evidence. The submission provides a clear formulation of adaptive rank control for robust reinforcement learning under epistemic uncertainty, together with NTK-regime analysis motivating a bias--variance tradeoff induced by rank selection. While the theoretical guarantees rely on local linearization/NTK-style assumptions and therefore do not fully characterize nonlinear deep RL training, the authors have clarified these limitations and the intended scope of the analysis.

The empirical evaluation is also sufficient for the claims made in the paper. Reviewers initially raised concerns about worst-case robustness, held-out parameterizations, performance drops after rank adaptation, the practical interpretation of sampling from a Wasserstein uncertainty set, and the mismatch between smooth-activation assumptions and ReLU-based experiments. The authors addressed these points in the rebuttal and revised manuscript by clarifying the distinction between representation-level robustness and worst-case robust control, adding/clarifying held-out and ablation experiments, explaining the replay-buffer-related source of transient performance drops, discussing the parametric approximation of the ambiguity set, and adding smooth-activation experiments. Overall, the evidence is adequate and convincing for the paper’s stated contribution.